# BREAKING THE DETECTION-GENERALIZATION PARADOX ON OUT-OF-DISTRIBUTION DATA

## ABSTRACT

The detection and generalization of out-of-distribution (OOD) data are critical in numerous real-world applications. While OOD detection enhances model reliability against outliers, generalization enables adaptability to unforeseen variations. Despite their importance, the intrinsic relationship between OOD detection and generalization remains underexplored, posing risks in practical deployments. This study identifies the *Detection-Generalization Paradox* on OOD data, where optimizing one objective can degrade the other. We investigate this paradox by analyzing the behaviors of models trained under different paradigms, focusing on representation, logits, and loss across in-distribution, covariate-shift, and semantic-shift data. Based on our findings, we propose Distribution-Robust Sharpness-Aware Minimization (DR-SAM), an optimization framework that balances OOD detection and generalization. DR-SAM employs both in-distribution and semantic-shift data during training, using data augmentation on in-distribution data to simulate potential covariate-shift scenarios and computing perturbations on model parameters to enhance OOD generalization. By determining the worst-case gradient direction, the model's decision boundary is adjusted to better encompass covariate-shift samples. Empirical evaluations demonstrate that DR-SAM improves the detection of semantic-shift samples and enhances generalization for covariate-shift samples.

## 1 INTRODUCTION

The detection and generalization of out-of-distribution (OOD) data are pivotal in real-world applications across various domains (Arjovsky, 2020; Salehi et al., 2021; Yang et al., 2022; 2024). Typical examples can be found in autonomous systems, medical diagnosis, and financial fraud detection (Sinha et al., 2022; Hilal et al., 2022; Hong et al., 2024). These aspects underscore the reliability and adaptability of machine learning models when exposed to multifarious data distributions.

In general, detecting OOD instances ensures reliability, while generalization empowers models to adapt to unforeseen OOD variations, as illustrated in Fig. 1. Methods of OOD detection utilize the model's probabilities or representations to identify OOD samples (Hendrycks et al., 2018; Lee et al., 2018; Liu et al., 2020; Zhang et al., 2023a). On the other side, the OOD generalization techniques, *e.g.*, regularization (Arjovsky et al., 2019; Krueger et al., 2021) and data augmentation (Kim et al., 2021; Hendrycks et al., 2021), facilitate the acquisition towards invariant representations. Notably, despite the real-world application *simultaneously* demanding both capabilities of detection and generalization, *the inherent relationship* between these two lines of research has yet to be elucidated comprehensively (Katz-Samuels et al., 2022; Bai et al., 2023). Such a knowledge gap entails hidden risks in the context of real-world applications.

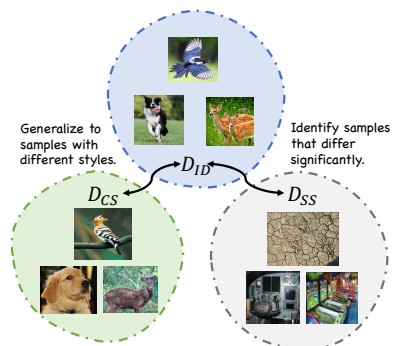

Figure 1: Data samples with semantic shift ($D_{SS}$) and covariate shift ($D_{CS}$) *w.r.t.* the in-distribution samples ($D_{ID}$).

In this study, we discover the underlying detection-generalization trade-off in OOD data arising from the prevailing detection (OOD-D) and generalization (OOD-G) methods, as shown in Fig. 2. We termed this trade-off as *Detection-Generalization Paradox*, namely, solely optimizing one objective with the corresponding method will degenerate the other. Nonetheless, optimizing the trade-off between these two essential targets remains under-explored and challenging (Bai et al., 2023; Katz-Samuels et al., 2022). Thus, we raise an open problem: *how to break the detection-generalization paradox to attain advanced abilities in both OOD detection and generalization simultaneously?*

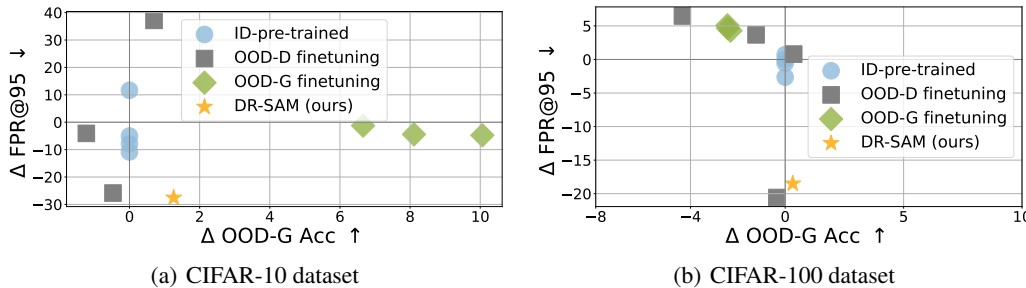

(a) CIFAR-10 dataset          (b) CIFAR-100 dataset

Figure 2: A performance comparison on both tasks of OOD detection and generalization. The ID-pre-trained models are finetuned with various OOD-D/OOD-G methods. The *variations* in OOD-D/OOD-G performance are calculated with metrics FPR@95/accuracy compared to the MSP baseline, where a *lower* $\Delta$FPR@95 indicates *improved* OOD-D and a *higher* $\Delta$Acc denotes *improved* OOD-G. As can be seen, OOD-D methods improve OOD-D but deteriorate OOD-G, while OOD-G methods enhance OOD-G but degenerate OOD-D. In contrast, DR-SAM improves both OOD-D and OOD-G.

To figure out the reason behind the paradox, we delve into the distinct behaviors when model training with different paradigms that induce the generalization-detection paradox. We start from the model's inference pipeline: $\forall (x, y) \sim D, \; x \xmapsto{f_{\boldsymbol{\theta}}^{\text{EMB}}(\cdot)} \boldsymbol{h}_x \xmapsto{f_{\boldsymbol{\theta}}^{\text{CLS}}(\cdot)} \hat{\boldsymbol{y}} \xmapsto{\mathcal{L}(\cdot, \cdot)} \mathcal{L}(\hat{\boldsymbol{y}}, y)$, in which we conduct an in-depth analysis of the three informative variables - representation $\boldsymbol{h}_x$, logits $\hat{\boldsymbol{y}}$, and loss $\mathcal{L}(\hat{\boldsymbol{y}}, y)$ with in-distribution data $D_{\text{ID}}$, covariate shift data $D_{\text{CS}}$, and semantic shift data $D_{\text{SS}}$. Specifically,

**Analysis on the representation space $\boldsymbol{h}_x$.** The OOD-D method enlarges the semantic gap between $D_{\text{ID}}$ and $D_{\text{SS}}$ but also enlarges the gap between $D_{\text{ID}}$ and $D_{\text{CS}}$. This explains its improvement in OOD-D and degeneration in OOD-G. On the other hand, the OOD-G method reduces the gap between $D_{\text{ID}}$ and $D_{\text{CS}}$ but also reduces the gap between $D_{\text{ID}}$ and $D_{\text{SS}}$. This explains its improvement in OOD-G and degeneration in OOD-D. Namely, neither line of the method can obtain the ideal representations.

**Analysis on the logit space $\hat{\boldsymbol{y}}$.** The OOD-D method enlarges the gap of prediction confidence between $D_{\text{ID}}$ and $D_{\text{SS}}$ (better OOD-D) and decreases the model's prediction on $D_{\text{CS}}$ (worse OOD-G). On the other side, the OOD-G method increases the confidence on both $D_{\text{CS}}$ (better OOD-G) and $D_{\text{SS}}$ (over-confident, worse OOD-D). These discoveries are aligned with those in representation space.

**Analysis on the loss space $\mathcal{L}(\hat{\boldsymbol{y}}, y)$.** We adopt the loss landscape and sharpness to investigate the flatness around the model's convergent point. Notably, the OOD-D method leads to a relatively flat landscape on $D_{\text{ID}}^{\text{test}}$ but with a sharper landscape on $D_{\text{CS}}^{\text{test}}$, indicating its degeneration in OOD generalization. Conversely, OOD-G method induces a flat landscapes on $D_{\text{ID}}^{\text{train}}$ and $D_{\text{CS}}^{\text{test}}$ but with a high sharpness on $D_{\text{ID}}^{\text{test}}$. An ideal model should possess low sharpness on both in-distribution and covariate-shift data. However, this cannot be achieved by the current OOD-D or OOD-G method.[1] We provide a detailed discussion of the sharpness of different methods in Sec. 3.2.

To break the paradox based on the above discoveries, one solution is to actively seek local flatness during training to enhance OOD generalization ability under the constraint of the OOD detection objective. Considering both detection and generalization objectives, we propose a novel optimization framework, Distribution-Robust Sharpness-Aware Minimization (DR-SAM).

Specifically, DR-SAM utilizes both in-distribution data and semantic-shift data in model training. The adoption of auxiliary semantic-shift outliers is to guarantee better OOD detection capability on test-time outliers encountered. Notably, in computing the perturbation on model parameters, we apply data augmentation on in-distribution data, aiming to simulate the potential covariate-shift data and thereby guarantee the model's OOD generalization capacity during the optimization procedure. Then, by applying an optimizer like the stochastic gradient descent (SGD), we obtain the gradient direction that indicates the worst-case from the current point. We can then extrapolate the decision boundary by shifting the current point towards this gradient direction to cover more covariate-shift samples.

Empirically, we evaluate DR-SAM on a series of OOD-D benchmarks with auxiliary OOD-G metrics. Extensive evaluations demonstrate that our method can achieve better OOD detection capability

---

[1]For clarity, our discussion here about the OOD-G methods does not include those methods that apply SAM or consider the mixup augmentation, as they inherently have the sharpness minimization effect during training.

for semantic-shift samples by lowering FPR@95 and improving generalization by increasing the classification accuracy on covariate-shift samples. It achieves up to a $9.82\%$ improvement in OOD-D and $7.02\%$ in OOD-G compared to the best baseline approaches. Further, we discuss the proposed method from different perspectives and provide a broad range of ablation studies and visualizations.

We summarize our main contributions as follows:

- We identify the detection-generalization paradox on out-of-distribution data among the prevailing OOD detection/generalization methods. Furthermore, we conduct an in-depth analysis to provide several insights into the paradox's intuitive manifestation and the underlying cause (Sec. 3).

- We propose a new optimization framework, Distribution-Robust Sharpness-Aware Minimization (DR-SAM), to break the detection-generalization paradox. By balancing OOD detection and generalization objectives, DR-SAM actively seeks local flatness to enhance OOD generalization ability under the constraint of OOD detection objective to guarantee dual capabilities (Sec. 4).

- Empirically, we conduct extensive experiments and justify that DR-SAM achieves leading performance in both measurements of OOD detection and generalization. We conduct several ablation studies and visualizations to provide further insights into the effectiveness of DR-SAM (Sec. 5).

## 2 PRELIMINARIES

**Notations.** We denote $D_{\text{ID}}$ as the **in-d**istribution data, $D_{\text{CS}}$ as the **c**ovariate **s**hift data for OOD generalization, and $D_{\text{SS}}$ as the **s**emantic **s**hift data for OOD detection. A sample $(x, y) \sim D$ contains image $x$ and label $y$. Besides, $f_{\boldsymbol{\theta}}(\cdot)$ denotes a classification model parameterized by $\boldsymbol{\theta}$. Here, $f_{\boldsymbol{\theta}}(\cdot) = f_{\boldsymbol{\theta}}^{\text{EMB}}(\cdot) \circ f_{\boldsymbol{\theta}}^{\text{CLS}}(\cdot)$, wherein the embedding module $f_{\boldsymbol{\theta}}^{\text{EMB}}(\cdot)$ encodes input $x$ to representation $\boldsymbol{h}_x$, and the following classification module $f_{\boldsymbol{\theta}}^{\text{CLS}}(\cdot)$ projects $\boldsymbol{h}_x$ to the classification logits $\hat{\boldsymbol{y}}$.

**OOD Detection (OOD-D)** aims to *identify* the semantic shift data $D_{\text{SS}}$ from in-distribution $D_{\text{ID}}$. Existing methods can be generally divided into two categories, *i.e.*, post-hoc approaches and finetuning approaches. Specifically, based on a pre-trained model, the post-hoc methods (Hendrycks & Gimpel, 2016; Lee et al., 2018; Liu et al., 2020; Zhou et al., 2021) adopt a score function $s(\cdot)$, *e.g.*, maximum softmax probability (MSP) or Mahalanobis distance, to project an input instance $x$ to a real value $s(x) \in \mathbb{R}$. The post-hoc score function $s(\cdot)$ is expected to maximize the gap between $D_{\text{ID}}$ and $D_{\text{SS}}$:

$$\max \mathbb{E}_{(x,y) \sim D_{\text{ID}}} \mathbb{E}_{(x',y') \sim D_{\text{SS}}} \left| s(x) - s(x') \right|. \tag{1}$$

Besides, fine-tuning approaches (Hendrycks et al., 2018; Du et al., 2022; Wang et al., 2023; Zhang et al., 2023a) further tune the trainable parameters to enhance the pre-trained model's OOD-D capability. Among these, the epidemic method of Outlier Exposure (OE) (Hendrycks et al., 2018) leverages *extra, exposures* OOD samples $D_{\text{SS}}^{\text{train}}$ during training, namely, learning with the objective:

$$\boldsymbol{\theta}^* = \min_{\boldsymbol{\theta}} \mathbb{E}_{(x,y) \sim D_{\text{ID}}^{\text{train}}} \mathcal{L}_{\text{CE}}(f_{\boldsymbol{\theta}}(x), y) + \lambda_{\text{OE}} \cdot \mathbb{E}_{(x',y') \sim D_{\text{SS}}^{\text{train}}} \mathcal{L}_{\text{OE}}(f_{\boldsymbol{\theta}}(x')), \tag{2}$$

where the $\mathcal{L}_{\text{OE}}$ forces the outlier's logits $f(x')$ to be close to a uniform distribution. OE helps the model to recognize characteristics that are specific to the training data $D_{\text{ID}}^{\text{train}}$ and generalize to the outliers during testing ($D_{\text{SS}}^{\text{test}}$). Empirically, OE is proven effective in enlarging the gap in Eqn. 1.

**OOD Generalization (OOD-G)** aims to *generalize* to the covariate shift data $D_{\text{CS}}$, which is achieved by optimizing models that have consistent performance across domains with different covariate shifts. Following (Arjovsky et al., 2019; Gulrajani & Lopez-Paz, 2021), we consider the dataset $D_e := \{(x_i^e, y_i^e)\}_{i=1}^{n_e}$ collected under multiple environment $e \in \mathcal{E}$. $D_{CS} = \{D_e : e \in \mathcal{E}\}$ denotes the collection of datasets from multiple environments with potential covariant shifts. The objective of OOD-G is given as:

$$\boldsymbol{\theta}^* = \min_{\boldsymbol{\theta}} \mathbb{E}_{e \in \mathcal{E}} \mathbb{E}_{(x,y) \sim D_e} \mathcal{L}_{\text{CE}}(f_{\boldsymbol{\theta}}(x), y) \text{ s.t. } \boldsymbol{\theta} \in \arg\min_{\boldsymbol{\theta}} \mathbb{E}_{(x',y') \sim D_e} \mathcal{L}_{\text{CE}}(f_{\boldsymbol{\theta}}(x'), y'), \tag{3}$$

To achieve the objective, IRM (Arjovsky et al., 2019) penalizes the model on domains with sub-optimal performance, while VREx (Krueger et al., 2021) optimizes the loss variance across domains.

**Sharpness-Aware Minimization (SAM)** (Foret et al., 2021) In general, optimizing neural networks minimizes target loss $\mathcal{L}$ by gradient descent. However, such an approach often causes the model to

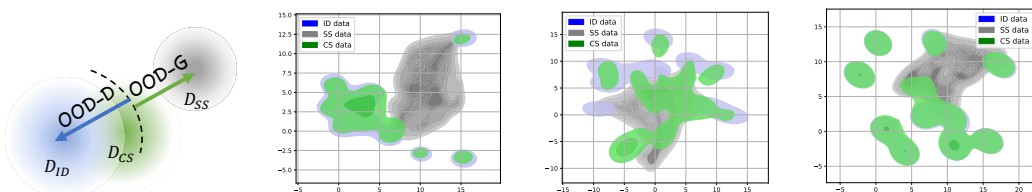

(a) Fine-tuning effects of OOD-D/OOD-G methods

(b) The ID-pre-trained model without fine-tuning

(c) Model fine-tuned with OOD-D method (Eqn. 2)

(d) Model fine-tuned with OOD-G method (Eqn. 3)

Figure 3: A comparison in the representation space. The effects of different training paradigms are illustrated in **(a)**. Specifically, compared to the pretrained model **(b)**, OOD-D method **(c)** enlarges the gap between $D_{\text{ID}}$ and $D_{\text{SS}}$, while OOD-G method **(d)** reduces the gap between $D_{\text{ID}}$ and $D_{\text{CS}}$.

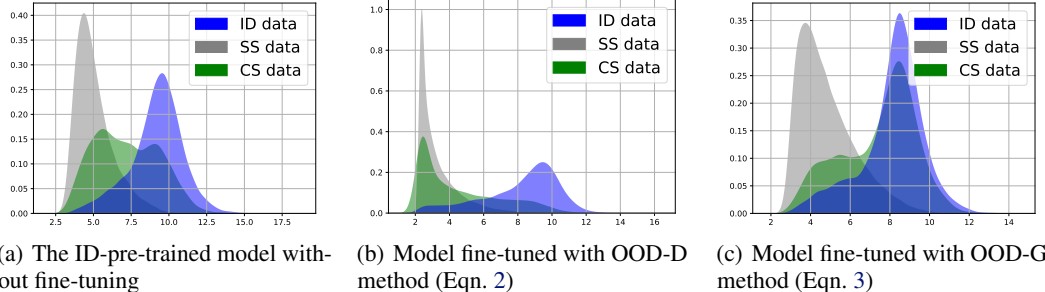

(a) The ID-pre-trained model without fine-tuning

(b) Model fine-tuned with OOD-D method (Eqn. 2)

(c) Model fine-tuned with OOD-G method (Eqn. 3)

Figure 4: A comparison of energy-based score (Liu et al., 2020) in the logit space. The gap in the embedding space is also reflected in the model's prediction. Specifically, OOD-D method **(b)** lowers the confidence of $D_{\text{CS}}$ and $D_{\text{SS}}$, while OOD-G method **(c)** increases the confidence of $D_{\text{CS}}$ and $D_{\text{SS}}$.

fall into a sharp local minimum. This leads the model to be sensitive to distribution shift (Chaudhari et al., 2019) and thus fails to generalize. To solve this, SAM is proposed to find a flat landscape within radius $\rho$ center at parameter $\boldsymbol{\theta}$ with the following objective:

$$\boldsymbol{\theta}^* = \min_{\boldsymbol{\theta}} \max_{\|\epsilon\|_2 \leq \rho} \mathbb{E}_{(x,y) \sim D_{\text{ID}}} \mathcal{L}_{\text{CE}}(f_{\boldsymbol{\theta}+\epsilon}(x), y), \tag{4}$$

where $\epsilon$ is the perturbation on $\boldsymbol{\theta}$. Through the min-max optimization, SAM can induce a less-sharp convergence point $\boldsymbol{\theta}^*$ and thus improve the model's generalizability.

## 3 AN IN-DEPTH ANALYSIS OF THE DETECTION-GENERALIZATION PARADOX

In this section, we delve into the distinct behaviors when model training with different paradigms that induce the generalization-detection paradox. We start from the model's inference pipeline:

$$D \in \{D_{\text{ID}}, D_{\text{CS}}, D_{\text{SS}}\}, \forall (x, y) \sim D, \ x \xmapsto{f_{\boldsymbol{\theta}}^{\text{EMB}}(\cdot)} \boldsymbol{h}_x \xmapsto{f_{\boldsymbol{\theta}}^{\text{CLS}}(\cdot)} \hat{\boldsymbol{y}} \xmapsto{\mathcal{L}(\cdot,\cdot)} \mathcal{L}(\hat{\boldsymbol{y}}, y). \tag{5}$$

In particular, we investigate the three informative variables, *i.e.*, representation $\boldsymbol{h}_x$, logits $\hat{\boldsymbol{y}}$, and loss $\mathcal{L}(\hat{\boldsymbol{y}}, y)$. Specifically, we conduct 1) a data-perspective analysis via the $\boldsymbol{h}_x$ and $\hat{\boldsymbol{y}}$ in Sec. 3.1, and 2) a model-perspective analysis via the landscape and sharpness on $\mathcal{L}(\hat{\boldsymbol{y}}, y)$ in Sec. 3.2. These two perspectives provide insights into the paradox's intuitive manifestation and underlying cause.

### 3.1 DATA-PERSPECTIVE ANALYSIS VIA REPRESENTATIONS AND LOGITS

**Analysis on the representation space.** An ideal model should possess (1) a small representation gap between $D_{\text{ID}}$ and $D_{\text{CS}}$, in order to successfully generalize to $D_{\text{CS}}$ samples, and (2) a large gap between $D_{\text{ID}}$ and $D_{\text{SS}}$ to clearly discriminate the $D_{\text{SS}}$ samples. However, this *cannot* be achieved by adopting the current OOD-D / OOD-G methods. As shown in Fig. 3, OOD-D method enlarges the gap between $D_{\text{ID}}$ and $D_{\text{SS}}$ (enhance the OOD-D) but also enlarges the gap between $D_{\text{ID}}$ and $D_{\text{CS}}$ (degenerate the OOD-G). On the other hand, OOD-G method reduces the gap between $D_{\text{ID}}$ and $D_{\text{CS}}$ (enhance the OOD-G) but also reduces the gap between $D_{\text{ID}}$ and $D_{\text{SS}}$ (degenerate the OOD-D).

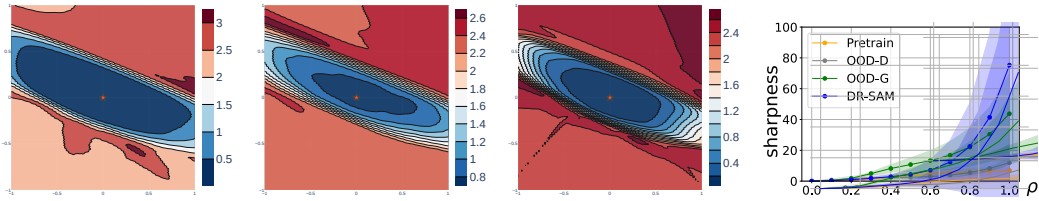

(a) Loss landscape of ID-pre-trained model  (b) Loss landscape of OOD-D fine-tuned model  (c) Loss landscape of OOD-G fine-tuned model  (d) Impact of fine-tuning on model's loss sharpness

Figure 5: A comparison in the loss space. We show the loss landscape and sharpness comparison across different training strategies. **(a)-(c)**: the loss landscapes of model training with different approaches on $D_{\text{ID}}^{\text{train}}$ using cross-entropy loss. **(d)**: sharpness comparison across different approaches.

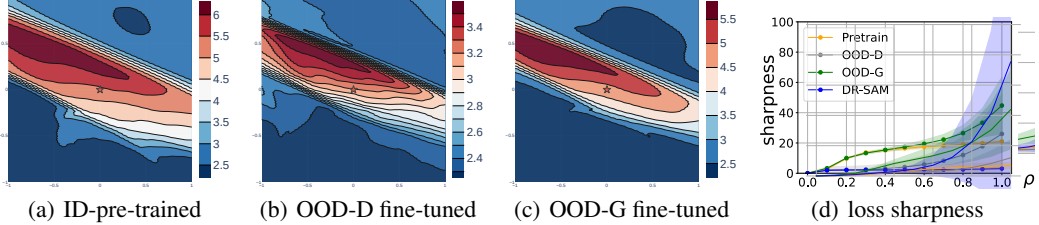

(a) ID-pre-trained  (b) OOD-D fine-tuned  (c) OOD-G fine-tuned  (d) loss sharpness

Figure 6: A comparison of loss landscape and sharpness across different training strategies on $D_{\text{ID}}^{\text{test}}$.

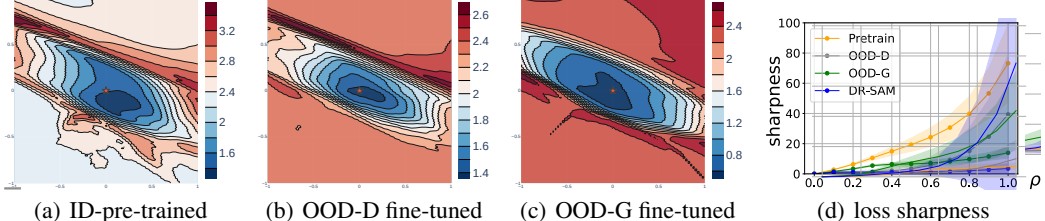

(a) ID-pre-trained  (b) OOD-D fine-tuned  (c) OOD-G fine-tuned  (d) loss sharpness

Figure 7: A comparison of loss landscape and sharpness across different training strategies on $D_{\text{CS}}^{\text{test}}$.

**Analysis on the logit space.** We employ the energy-based score in Eqn. 1 to indicate the model prediction confidence. As shown in Fig. 4, the OOD-D method enlarges the gap between $D_{\text{ID}}$ and $D_{\text{CS}}$ would decrease the model's prediction on $D_{\text{CS}}$ (degenerate the OOD-G). On the other side, the OOD-G would increase the confidence on both $D_{\text{SS}}$ and $D_{\text{CS}}$ (degenerate the OOD-D).

## 3.2 MODEL-PERSPECTIVE ANALYSIS VIA LOSS SHARPNESS AND LANDSCAPE

Next, we investigate from the lens of landscape (Li et al., 2018a) and sharpness (Keskar et al., 2016). **Landscape** indicates the flatness of local minima $\boldsymbol{\theta}$. Specifically, with a chosen center point $\boldsymbol{\theta}$ and two direction vectors, $\boldsymbol{\delta}$ and $\boldsymbol{\eta}$, we plot the 2D surface landscape using the following function:

$$\text{landscape}(\boldsymbol{\theta}, D, \alpha, \beta) = \mathbb{E}_{(x,y)\sim D}\mathcal{L}(f_{\boldsymbol{\theta}+\alpha\boldsymbol{\delta}+\beta\boldsymbol{\eta}}(x), y). \tag{6}$$

**Shaprness** also indicates the flatness of the converging area around parameters $\boldsymbol{\theta}$. Given a model $f_{\boldsymbol{\theta}}$ on a dataset $D$ and loss function $\mathcal{L}$, the sharpness of the region at radii $\rho$ center at $\boldsymbol{\theta}$ is given as:

$$\text{sharpness}(\boldsymbol{\theta}, D) \triangleq \max_{\|\epsilon\|_2 \leq \rho} \mathbb{E}_{(x,y)\sim D}\big[\mathcal{L}(f_{\boldsymbol{\theta}+\epsilon}(x), y) - \mathcal{L}(f_{\boldsymbol{\theta}}(x), y)\big], \tag{7}$$

where $\epsilon$ is small perturbation imposing on $\boldsymbol{\theta}$. A lower sharpness value indicates less loss variance within radii $\rho$ and, thereby, a flatter neighborhood around the converging area of $\boldsymbol{\theta}$. A flatter region around $\boldsymbol{\theta}$ generally leads the model to have a better generalization ability (Foret et al., 2021).

Here, we visualize the model's loss landscape and sharpness in Figs. 5, 6 and 7, wherein the model witnesses different landscapes when various distributions and training methods. Overall, the qualitative landscapes (Figs. 5(a-c)) are aligned with the quantitative sharpness curves (Fig. 5(d)).

Notably, OOD-D method leads to a relatively flat landscape on the $D_{\text{ID}}^{\text{test}}$ but experiences large sharpness in the $D_{\text{CS}}^{\text{test}}$, indicating its degeneration in generalization ability. Conversely, the OOD-G

fine-tuned model has flat landscapes (low sharpness) on $D_{\text{ID}}^{\text{train}}$ and $D_{\text{CS}}^{\text{test}}$ but with a high sharpness on $D_{\text{ID}}^{\text{test}}$. Namely, this model generalizes well on $D_{\text{CS}}^{\text{test}}$ at the cost of worse performance on $D_{\text{ID}}^{\text{test}}$.

Therefore, an ideal model should have low sharpness on both in-distribution and covariate-shift data. However, this cannot be guaranteed by solely adopting the current OOD-D or OOD-G method that optimizes the loss via typical optimization methods like SGD or Adam, as it might fall into the locally sharp minima that frequently exist in the complex high-dimensional space (Keskar et al., 2016; Garipov et al., 2018; Izmailov et al., 2019; Foret et al., 2021; Dziugaite & Roy, 2017; Jiang et al., 2019; Cha et al., 2021). Note that, the range of OOD-G methods that we discussed here does not include those methods that apply SAM or consider the mixup augmentation, as they inherently have the sharpness minimization effect during training. Besides, it is discovered that a lower sharpness on $D_{\text{ID}}^{\text{train}}$ might ensure a lower sharpness on $D_{\text{ID}}^{\text{test}}$ or $D_{\text{CS}}^{\text{test}}$. However, a steady low sharpness across a range of $\rho$ on $D_{\text{ID}}^{\text{train}}$ might not ensure low sharpness on $D_{\text{ID}}^{\text{test}}$ and $D_{\text{CS}}^{\text{test}}$. We refer to Appendix. C for further discussions. To handle the paradox, one conceptual idea is to actively seek flatness within radii $\rho$ to enhance generalization ability under the constraint of OOD-D objective to guarantee the detection capability.

## 4  DR-SAM: DISTRIBUTION-ROBUST SHARPNESS-AWARE MINIMIZATION

Recall that the generalization ability is bounded by the neighborhood-wise training loss (Foret et al., 2021), which is indicated by the sharpness. Taking both objectives of OOD detection and generalization into consideration, we propose a novel optimization framework, Distribution-Robust Sharpness-Aware Minimization (DR-SAM). The overall pipeline of the proposed method is elaborated in Algorithm 1. Specifically, DR-SAM obtains the optimal parameters $\boldsymbol{\theta}^*$ as follows:

$$\boldsymbol{\theta}^* = \min_{\boldsymbol{\theta}} \max_{\|\epsilon\|_2 \leq \rho} \underbrace{\mathbb{E}_{x \sim D_{\text{ID}}^{\text{train}}} \mathcal{L}_{\text{CE}}(f_{\boldsymbol{\theta}+\epsilon}(x), y) + \lambda \cdot \mathbb{E}_{x' \sim D_{\text{SS}}^{\text{train}}} \mathcal{L}_{\text{OE}}(f_{\boldsymbol{\theta}+\epsilon}(x'))}_{\mathcal{L}_{\text{DR-SAM}}(f_{\boldsymbol{\theta}+\epsilon}, D_{\text{ID}}^{\text{train}}, D_{\text{SS}}^{\text{train}})}, \quad (8)$$

where $\epsilon$ denotes the perturbation on model parameters. Notably, the $\epsilon$ here is calculated as:

$$\epsilon = \frac{\rho \cdot \nabla_{\boldsymbol{\theta}} \mathcal{L}_{\text{DR-SAM}}(f_{\boldsymbol{\theta}+\epsilon}, \texttt{aug}(D_{\text{ID}}^{\text{train}}), D_{\text{SS}}^{\text{train}})}{||\nabla_{\boldsymbol{\theta}} \mathcal{L}_{\text{DR-SAM}}(f_{\boldsymbol{\theta}+\epsilon}, \texttt{aug}(D_{\text{ID}}^{\text{train}}), D_{\text{SS}}^{\text{train}})||}, \quad (9)$$

where the augmentation $\texttt{aug}(\cdot)$ on in-distribution data aims to simulate the covariate-shift data and guarantee the model's OOD generalization capacity during optimization. Then, by applying an optimizer, *e.g.*, the stochastic gradient descent (SGD), we obtain the gradient direction of Eqn. 9 that indicates the *worst-case* from the current point. By shifting $\boldsymbol{\theta}$ towards this gradient direction, we can extrapolate the decision boundary with ratio $\rho$ to cover more covariate-shift samples. Besides, the adoption of auxiliary outliers $D_{\text{SS}}^{\text{train}}$ is to guarantee better OOD detection capability on test-time outliers encountered. Then, the parameters are iteratively updated with learning rate $\eta$ as:

$$\boldsymbol{\theta}' = \boldsymbol{\theta} - \eta \nabla_{\boldsymbol{\theta}+\epsilon} \mathcal{L}_{\text{DR-SAM}}(f_{\boldsymbol{\theta}+\epsilon}, D_{\text{ID}}^{\text{train}}, D_{\text{SS}}^{\text{train}}). \quad (10)$$

---

**Algorithm 1** DR-SAM: Distribution-Robust Sharpness-Aware Minimization

---

**Require:** In-distribution data $D_{\text{ID}}^{\text{train}}$, auxiliary semantic-shift data $D_{\text{SS}}^{\text{train}}$, data augmentation operator $\texttt{aug}$, an ID-pre-trained model $f_{\boldsymbol{\theta}}$, number of iterations $T$.

1: **for** $t = 1 \ldots T$ **do**
2:     Sample mini-batch data $B_{\text{ID}} \subset D_{\text{ID}}^{\text{train}}$ and $B_{\text{SS}} \subset D_{\text{SS}}^{\text{train}}$.
3:     Compute the loss $\mathcal{L}_\epsilon \leftarrow \mathcal{L}_{\text{DR-SAM}}(f_{\boldsymbol{\theta}}, \texttt{aug}(B_{\text{ID}}^{\text{train}}), B_{\text{SS}}^{\text{train}})$.
4:     Compute the perturbation $\epsilon \leftarrow \frac{\rho \cdot \nabla_{\boldsymbol{\theta}} \mathcal{L}_\epsilon}{||\nabla_{\boldsymbol{\theta}} \mathcal{L}_\epsilon||}$.
5:     Compute the loss $\mathcal{L}_{\theta+\epsilon} \leftarrow \mathcal{L}_{\text{DR-SAM}}(f_{\boldsymbol{\theta}+\epsilon}, B_{\text{ID}}^{\text{train}}, B_{\text{SS}}^{\text{train}})$.
6:     Update the parameters $\boldsymbol{\theta} \leftarrow \boldsymbol{\theta} - \eta \nabla_{\boldsymbol{\theta}+\epsilon} \mathcal{L}_{\theta+\epsilon}$.
7: **end for**
8: **return** The fine-tuned model $f_{\boldsymbol{\theta}}$.

---

It is noteworthy that the most distinct point in DR-SAM compared with the vanilla SAM is the $\texttt{aug}(\cdot)$, which is applied solely during the perturbation generation stage. The intuition here is to create a challenging perturbation aware of the worst case for generalization under covariate shift, rather than only exploring the separation between in-distribution data and out-of-distribution data. The reason

that we go back to the training without $\text{aug}(\cdot)$ at the parameter optimization stage (see Lines 5 and 6 in Algorithm 1) is to preserve the benefits of the training trajectory for OOD-D, enabling us to simultaneously improve the generalization and detection in a more harmonious manner. As will be shown later, such a novel design brings us a significant gain compared with the vanilla SAM in the following experiments, which helps to break the generalization-detection paradox for OOD data.

## 4.1 COMPARING DR-SAM WITH THE MOST-RELATED WORKS

Several previous works also study the model's detection-generalization problem. The pioneer work SCONE (Bai et al., 2023) proposes to enhance the model's OOD-D and OOD-G ability simultaneously when training the model with wild data. In contrast, DR-SAM follows the common practice in OOD-D. Unlike SCONE, DR-SAM employs training distribution containing only labeled in-distribution samples and unlabeled auxiliary outliers. In addition, DR-SAM focuses on alleviating the generalization trade-off due to the employment of auxiliary outliers. Whereas SCONE (Bai et al., 2023) pays more attention to handling samples from mixing types of distributions. Averly & Chao (2023) proposes a novel evaluation framework that focuses on detecting and rejecting samples beyond the model's classification capability, which aims to evaluate the existing model's robustness with samples from mixing distribution types. Conversely, DR-SAM simultaneously enhances the model's generalization and detection ability.

The most related work DUL (Zhang et al., 2024) propose a novel and theoretical guarantee optimization framework to enhance the model's OOD-D ability while maintaining the original OOD-G capability. DUL incorporates distributional uncertainty in the Bayesian framework to bridge the detection and generalization learning target. Specifically, DUL encourages the exposed outlier to have high uncertainty while maintaining a non-increased overall uncertainty to ensure generalization capability. Instead of focusing on decoupling the uncertainty under the Bayesian framework, we propose DR-SAM to minimize the model's loss of sharpness under the framework of SAM. We provide a detailed discussion in Appendix G.3.

## 5 EXPERIMENTS

In this section, we present the comprehensive experiments of the proposed method. To begin with, we provide the experimental setups in detail in Sec. 5.1. Next, we compare the proposed method's performance with a series of post-hoc scoring functions and the OE-based approaches with different strategies on the adopted auxiliary outliers in Sec. 5.2. Then, we conduct various ablation studies in Sec. 5.3 and visualizations in Sec. 5.4 to provide further insights into the proposed method.

## 5.1 SETUPS

To evaluate the proposed method under complex real-world scenarios, we employ a wide range of datasets that cover different levels of distribution shift $w.r.t.$ the in-distribution data $D_{\text{ID}}$, which are listed as follows. All experiments are conducted on ResNet-18 that pre-trained on respecting $D_{\text{ID}}$.

**CIFAR Benchmark.** We empirically evaluate the proposed method on CIFAR-10 and CIFAR-100 benchmarks based on OpenOOD (Zhang et al., 2023b). For the auxiliary outliers, we adopt the TIN-597 (Zhang et al., 2023b) that sampled from ImageNet-1K (Deng et al., 2009) and has no overlap with the test sets. For CIFAR-10/100, we consider TinyImageNet (Le & Yang, 2015) and CIFAR-100/10 as Near-OOD considering their semantic similarity to the $D_{\text{ID}}$. CIFAR-10/100 share the same group of Far-OOD datasets, namely MNIST (Deng, 2012), SVHN (Netzer et al., 2011), Textures (Cimpoi et al., 2014), and Places365 (Zhou et al., 2018). To assess the model's generalization ability, we employ the covariate-shifted CIFAR datasets (Hendrycks & Dietterich, 2018), namely CIFAR-10-C and CIFAR-100-C, which adopt various image augmentation with different strengths.

**ImageNet-200 Benchmark.** The ImageNet-200 is the subset of the ImageNet-1K (Deng et al., 2009). The rest of the samples act as the auxiliary outliers for training. We employ SSB-hard (Vaze et al., 2021) and NINCO (Bitterwolf et al., 2023) as Near-OOD. Far-OOD includes iNaturalist (Van Horn et al., 2018), Textures, and OpenImage-O (Wang et al., 2022). To assess the generalization ability, we adopt the ImageNet-R (Hendrycks et al., 2021) with style change from the $D_{\text{ID}}$.

| Method | Near-OOD | | Far-OOD | | Average | | Accuracy | |
|---|---|---|---|---|---|---|---|---|
| | FPR@95↓ | AUROC↑ | FPR@95↓ | AUROC↑ | FPR@95↓ | AUROC↑ | ID↑ | OOD↑ |
| Post-hoc | | | | | | | | |
| MSP | 48.14±4.02 | 88.04±0.26 | 31.68±1.77 | 90.75±0.43 | 39.91 | 89.39 | 95.06±0.30 | 79.24±0.45 |
| EBO | 61.63±4.91 | 87.61±0.47 | 41.47±5.09 | 91.25±0.90 | 51.55 | 89.43 | 95.06±0.30 | 79.24±0.45 |
| Gen | 53.85±3.10 | 88.22±0.30 | 34.60±1.42 | 91.38±0.68 | 44.23 | 89.80 | 95.06±0.30 | 79.24±0.45 |
| KNN | 33.91±0.38 | 90.67±0.20 | 24.21±0.45 | 92.99±0.14 | 29.06 | 91.83 | 95.06±0.30 | 79.24±0.45 |
| RMDS | 38.78±2.52 | 89.83±0.28 | 25.25±0.70 | 92.23±0.21 | 32.02 | 91.03 | 95.06±0.30 | 79.24±0.45 |
| VIM | 44.76±2.18 | 88.73±0.28 | 24.97±0.49 | 93.50±0.23 | 34.87 | 91.12 | 95.06±0.30 | 79.24±0.45 |
| Training Methods (w/o Outlier Data) | | | | | | | | |
| ConfBranch | 36.96±0.52 | 87.37±0.33 | 30.70±0.56 | 88.78±0.33 | 33.83 | 88.08 | 93.66±0.27 | 77.19±0.39 |
| G-ODIN | 63.32±5.43 | 84.35±1.38 | 49.54±6.31 | 88.54±1.37 | 56.43 | 86.45 | 93.33±0.25 | 77.56±0.24 |
| LogitNorm | 28.32±0.85 | 92.40±0.24 | 15.17±1.04 | 96.05±0.19 | 21.75 | 94.23 | 94.89±0.19 | 79.09±0.66 |
| Training Methods (w/ Outlier Data) | | | | | | | | |
| OE | 17.42±0.75 | 95.51±0.26 | 10.70±0.58 | 97.33±0.23 | 14.06 | 96.42 | 95.02±0.08 | 78.77±0.38 |
| MCD | 38.48±0.38 | 87.94±0.07 | 33.20±0.46 | 89.27±0.18 | 35.84 | 88.61 | 93.92±0.11 | 78.01±0.23 |
| MixOE | 87.53±6.17 | 85.19±1.78 | 66.63±12.85 | 89.11±1.89 | 77.08 | 87.15 | 96.13±0.18 | 79.94±0.58 |
| OOD-G Methods (w/ covariate shift samples) | | | | | | | | |
| ERM | 43.51±4.66 | 88.67±0.36 | 27.44±1.75 | 91.26±0.26 | 35.48 | 89.97 | 93.94±0.23 | 89.52±0.24 |
| VRE-x | 43.96±5.32 | 88.68±0.40 | 28.69±5.20 | 91.10±0.66 | 36.33 | 89.89 | 93.86±0.00 | 89.48±0.10 |
| GroupDRO | 42.37±4.79 | 88.75±0.31 | 27.92±4.24 | 91.31±0.56 | 35.15 | 90.03 | 93.84±0.20 | 89.30±0.21 |
| Sharpness-based Methods | | | | | | | | |
| SAM | 28.86±0.50 | 91.09±0.10 | 21.45±1.03 | 93.08±0.39 | 25.15 | 92.08 | 95.69±0.21 | 80.69±0.65 |
| **DR-SAM** | 18.81±1.07 | 95.41±0.20 | 5.90±1.39 | 98.67±0.40 | 12.36 | 97.04 | 95.13±0.19 | 80.50±0.55 |

Table 1: Performance comparison on the CIFAR-10 benchmark. All methods are trained on the same backbone. ↑ indicates larger values are preferred, and ↓indicates smaller values are better. All performances are reported by percentage and are averaged by multiple trials.

| Method | Near-OOD | | Far-OOD | | Average | | Accuracy | |
|---|---|---|---|---|---|---|---|---|
| | FPR95↓ | AUROC↑ | FPR95↓ | AUROC↑ | FPR95↓ | AUROC↑ | ID↑ | OOD↑ |
| Post-hoc | | | | | | | | |
| MSP | 54.84±0.58 | 80.21±0.13 | 58.52±1.12 | 77.83±0.45 | 56.68 | 79.02 | 77.26±0.09 | 36.59±0.04 |
| RMDS | 55.43±0.31 | 80.17±0.09 | 52.65±0.64 | 82.97±0.42 | 54.04 | 81.57 | 77.26±0.09 | 36.59±0.04 |
| Gen | 54.23±0.54 | 81.27±0.10 | 57.04±1.01 | 79.59±0.54 | 55.64 | 80.43 | 77.26±0.09 | 36.59±0.04 |
| EBO | 55.77±0.64 | 80.82±0.09 | 56.47±1.42 | 79.83±0.62 | 56.12 | 80.33 | 77.26±0.09 | 36.59±0.04 |
| VIM | 62.61±0.24 | 75.04±0.14 | 50.75±1.00 | 81.64±0.63 | 56.68 | 78.34 | 77.26±0.09 | 36.59±0.04 |
| KNN | 61.18±0.13 | 80.16±0.16 | 53.61±0.25 | 82.43±0.17 | 57.40 | 81.30 | 77.26±0.09 | 36.59±0.04 |
| Training Methods (w/o Outlier Data) | | | | | | | | |
| ConfBranch | 78.04±0.11 | 67.30±0.08 | 74.34±2.27 | 63.99±0.94 | 76.19 | 65.65 | 74.93±0.27 | 35.49±0.13 |
| G-ODIN | 68.07±6.02 | 74.72±2.93 | 56.42±2.87 | 78.80±1.74 | 62.25 | 76.76 | 69.87±4.46 | 33.10±2.11 |
| LogitNorm | 58.99±0.46 | 79.69±0.31 | 48.61±1.69 | 82.91±1.21 | 53.80 | 81.30 | 75.83±0.26 | 35.92±0.12 |
| Training Methods (w/ Outlier Data) | | | | | | | | |
| OE | 33.20±0.59 | 87.16±0.44 | 39.03±1.47 | 88.12±0.39 | 36.11 | 87.64 | 76.51±0.35 | 36.24±0.16 |
| MCD | 58.68±0.30 | 77.74±0.18 | 62.02±0.33 | 75.91±0.13 | 60.35 | 76.83 | 74.66±0.31 | 35.36±0.15 |
| MixOE | 56.89±2.04 | 80.44±0.69 | 58.06±4.45 | 78.70±2.03 | 57.48 | 79.57 | 78.01±0.05 | 36.95±0.02 |
| OOD-G Methods (w/ Covariate shift samples) | | | | | | | | |
| ERM | 59.84±0.15 | 77.89±0.09 | 63.67±0.83 | 73.13±0.14 | 61.76 | 75.51 | 72.10±0.29 | 34.15±0.14 |
| VRE-x | 59.22±0.53 | 77.99±0.05 | 63.49±1.29 | 73.52±0.35 | 61.35 | 75.76 | 72.14±0.49 | 34.17±0.23 |
| GroupDRO | 59.29±0.03 | 77.87±0.08 | 62.67±0.33 | 73.98±0.15 | 60.98 | 75.92 | 72.37±0.33 | 34.28±0.16 |
| Sharpness-based Methods | | | | | | | | |
| SAM | 52.32±0.15 | 81.32±0.07 | 55.50±0.19 | 78.97±0.33 | 53.91 | 80.14 | 78.53±0.27 | 37.20±0.13 |
| **DR-SAM** | 29.58±0.20 | 89.13±0.03 | 46.77±1.54 | 84.73±0.80 | 38.18 | 86.93 | 77.91±0.28 | 36.91±0.13 |

Table 2: Performance comparison on the CIFAR-100 benchmark. All methods are trained on the same backbone. ↑ indicates larger values are preferred, and ↓indicates smaller values are better.

**Metrics.** To assess the detection ability, we adopt (1) area under the receiver operating characteristic curve (AUROC), which reflects the probability that a positive sample scores higher than a negative one (Fawcett, 2006), and (2) false positive rate at 95% true positive rate (FPR@95) (Liang et al., 2017), which indicates the probability of misclassifying a negative sample as positive when the true positive rate is 95%. We also report the classification accuracy oon $D_{\text{ID}}$ and $D_{\text{CS}}$ for evaluation.

**Baselines.** We compare the proposed method with approaches that excel in OOD-D or OOD-G. We employ six post-hoc scoring functions, namely maximum softmax probability (MSP) (Hendrycks & Gimpel, 2016), energy score (EBO) (Liu et al., 2020), relative Mahalanobis distance (RMDS) (Ren et al., 2021), GEN (Liu et al., 2023), VIM (Wang et al., 2022), and KNN (Sun et al., 2022). For methods that do not require outliers, we adopt ConfBranch (DeVries & Taylor, 2018), G-ODIN (Hsu et al., 2020), and LogitNorm (Wei et al., 2022). For methods that require outliers, we employ OE (Hendrycks et al., 2018), MCD (Yu & Aizawa, 2019), and MixOE (Zhang et al., 2023a). For OOD-G, we consider ERM, VRE-x (Krueger et al., 2021), and GroupDRO (Sagawa et al., 2020).

| Method | Near-OOD | | Far-OOD | | Average | | Accuracy | |
|---|---|---|---|---|---|---|---|---|
| | FPR95↓ | AUROC↑ | FPR95↓ | AUROC↑ | FPR95↓ | AUROC↑ | ID↑ | OOD↑ |
| Post-hoc | | | | | | | | |
| MSP | 54.93±0.36 | 83.29±0.06 | 35.39±0.38 | 90.13±0.09 | 45.16 | 86.71 | 86.38±0.07 | 43.85±0.09 |
| RMDS | 54.09±0.64 | 82.54±0.26 | 32.50±0.77 | 88.06±0.35 | 43.30 | 85.30 | 86.38±0.07 | 43.85±0.09 |
| Gen | 55.16±0.13 | 83.64±0.04 | 32.13±0.64 | 91.35±0.11 | 43.65 | 87.50 | 86.38±0.07 | 43.85±0.09 |
| EBO | 60.31±0.56 | 82.41±0.07 | 34.84±1.29 | 90.84±0.22 | 47.58 | 81.63 | 86.38±0.07 | 43.85±0.09 |
| VIM | 59.36±0.75 | 78.59±0.24 | 27.22±0.34 | 91.26±0.18 | 43.29 | 84.93 | 86.38±0.07 | 43.85±0.09 |
| KNN | 60.32±0.50 | 81.46±0.18 | 27.30±0.73 | 93.15±0.23 | 43.81 | 87.31 | 86.38±0.07 | 43.85±0.09 |
| Training Methods (w/o Outlier Data) | | | | | | | | |
| ConfBranch | 61.56±0.43 | 79.00±0.14 | 33.56±0.41 | 90.64±0.05 | 47.56 | 84.82 | 86.23±0.23 | 44.34±0.26 |
| G-ODIN | 68.92±0.60 | 77.62±0.07 | 29.69±1.01 | 92.25±0.13 | 49.31 | 84.94 | 85.38±0.15 | 43.19±0.02 |
| LogitNorm | 57.36 ±0.57 | 82.23±0.15 | 26.61±0.52 | 92.92±0.25 | 41.99 | 87.58 | 86.11±0.27 | 44.32±0.34 |
| Training Methods (w/ Outlier Data) | | | | | | | | |
| OE | 52.44±0.67 | 86.93±0.21 | 36.75±1.31 | 87.88±0.31 | 44.60 | 87.41 | 85.67±0.43 | 43.30±0.40 |
| MCD | 59.60±0.63 | 79.05±0.12 | 44.48±0.75 | 86.08±0.21 | 52.04 | 82.57 | 79.33±0.34 | 35.09±0.16 |
| MixOE | 60.50±1.27 | 82.08±0.43 | 42.36±1.85 | 87.66±0.50 | 51.43 | 84.87 | 86.76±0.43 | 43.49±0.19 |
| Ours | | | | | | | | |
| DR-SAM | 52.23±0.60 | 85.10±0.21 | 34.01±0.69 | 89.02±0.19 | 43.12 | 87.06 | 86.63±0.15 | 46.93±0.12 |

Table 3: Performance comparison on the ImageNet-200 benchmark. All methods are trained on the same backbone. ↑ indicates larger values are preferred, and ↓indicates smaller values are better.

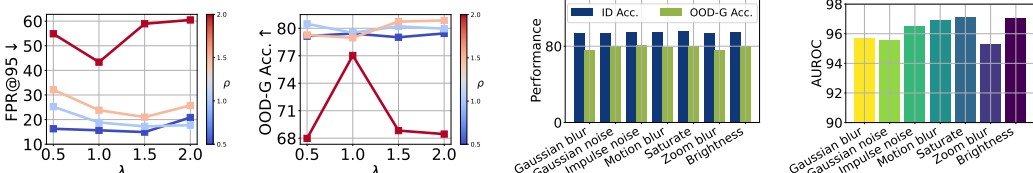

(a) Impact of perturbation strength on OOD-D (b) Impact of perturbation strength on OOD-G (c) Impact of different augmentation on accuracy (d) Impact of different augmentation on AUROC

Figure 8: Ablation study on perturbation factors of DR-SAM.

## 5.2 MAIN RESULTS

In this part, we present the comparison and discussion of different approaches to demonstrate the effectiveness of the proposed method. Since OE-based approaches involve auxiliary outliers during training, the model generally achieves improved empirical performance on detection-related metrics.

For the training methods, G-ODIN and MCD caused a 9.54% and 9.53%, respectively, decrease in OOD-G accuracy compared to the original MSP baseline as shown in Tab. 1 and Tab. 2. G-ODIN decomposes softmax confidence to separate semantic samples from $D_{ID}$, while MCD ensembles multiple classification heads to promote disagreement between each head's predictions on OOD samples. We attribute this decline to the explicitly promoted disagreement between $D_{ID}$ and $D_{SS}$, without adequately considering the influence of $D_{CS}$. LogitNorm, ConfBranch, and OE also suffer from a decrease in the ability to generalize. However, we observe that MixOE helps the model to generalize to covariate shift samples. By constructing mixed samples using $D_{ID}$ and auxiliary outliers for training, which could cover parts of $D_{CS}$ and improve the model's generalization ability.

We observe a positive fine-tuning with OOD-G approaches in Tab. 1. These methods enhance detection and generalization in both realms, though the improvement in detection is negligible compared to generalization. However, on more fine-grand datasets like CIFAR-100 (Tab. 2), the OOD-G methods suffer from up to 8% decrease in the FPR@95 compared to the MSP. Our method can consistently achieve better detection performance across the CIFAR benchmarks and shows scalability in the ImageNet-200 benchmark, as shown in Tab. 3. DR-SAM shows improvement in handling covariate shift samples, which verifies its effectiveness in addressing the trade-off. We provide further experiments and discussions in Appendix B.

## 5.3 ABLATION STUDIES

**The impact of $\rho$.** We aim to understand how radii $\rho$ impact the model's performance on OOD-D/G under different choices of $\rho$ and $\lambda$, shown in Fig. 8. Notably, a flat region with small $\rho$ generally enhances the performance of both OOD-D and OOD-G, as indicated by low FPR@95 in Fig. 8(a) and high accuracy in Fig. 8(b). However, a smaller $\rho$ is preferred for the OOD-D, while OOD-G pursues

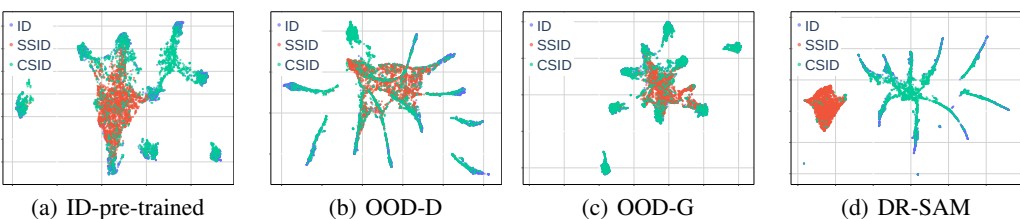

Figure 9: A comparison in the representation space for different methods.

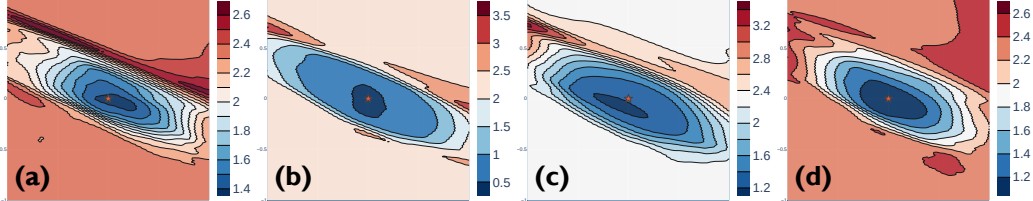

Figure 10: Loss landscape visualization for different methods on $D_{CS}^{test}$ of CIFAR-10. From (a) to (d): OE, ERM, SAM, and DR-SAM.

a larger flatness region around the coverage area. Recall the aforementioned analysis in Sec. 3, which indicates that the model should seek flatness within specific $\rho$ to boost OOD-D/G. We provide further experiments and comprehensive comparison with SAM in Appendix A, which shows the capability for OOD-G of DR-SAM is derived from data augmentation.

**The impact of aug$(\cdot)$.** In Fig. 8, we demonstrate the impact of different augmentation on the model's OOD-D/G performance. We employ different augmentations that are adopted in Hendrycks & Dietterich (2018). As shown in Fig. 8(c), different augmentations generally affect both in-distribution and covariate-shift samples, wherein Saturate and Brightness have the most positive impact on the model's generalization ability. In Fig. 8(d), we show the augmentation's impact on OOD-D performance. Generally, a proper augmentation can enhance both OOD-D and OOD-G simultaneously by helping the model to enlarge the gap between $D_{CS}$ and $D_{SS}$.

### 5.4 VISUALIZATION

In this part, we conduct further discussions from the lens of loss landscape and representation visualization. DR-SAM successfully achieves the previously mentioned desired property for the model to handle the detection-generalization paradox. In Fig. 9(d), we demonstrate that DR-SAM can enlarges the gap between $D_{ID}$ and $D_{SS}$ while reduce the gap between $D_{ID}$ and $D_{CS}$.

In Fig. 10, we demonstrate the loss of landscapes for different methods on $D_{CS}^{test}$ of CIFAR-10. OE leads the model to minima with high sharpness, making it sensitive to distribution shift, thus enhancing the model's detection ability while sacrificing the generalization performance. ERM achieves higher sharpness than SAM and DR-SAM, but it does not explicitly optimize the loss's sharpness. DR-SAM, on the other hand, achieves lower sharpness within the specific radii but high at the outer region. This allows the model to be robust to the covariate-shifted samples while sensitive to the semantic-shifted ones.

### 6 CONCLUSION

In this paper, we identify the Detection-Generalization Paradox on out-of-distribution (OOD) data, where optimizing for detection can degrade generalization and vice versa. We analyze this paradox by examining models trained under various paradigms, focusing on representation, logits, and loss across different data shifts. To address this, we propose Distribution-Robust Sharpness-Aware Minimization (DR-SAM), an optimization framework that balances OOD detection and generalization. DR-SAM uses data augmentation and perturbations on model parameters to simulate covariate-shift scenarios and enhance OOD generalization. Empirical results show that DR-SAM improves the detection of semantic-shift samples and boosts generalization for covariate-shift samples.

ETHIC STATEMENT

The study does not involve human subjects, data set releases, potentially harmful insights, applications, conflicts of interest, sponsorship, discrimination, bias, fairness concerns, privacy or security issues, legal compliance issues, or research integrity issues.

REPRODUCIBILITY STATEMENT

The experimental setups for training and evaluation are described in detail in Appendix F, and the experiments are all conducted using public datasets. We provide the link to our source codes to ensure the reproducibility of our experimental results: https://anonymous.4open.science/r/DR-SAM-9C89/.

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

APPENDIX

## A  COMPARING DR-SAM WITH SAM

(1) **A key difference between DR-SAM and the standard SAM is employing aug(·) only during the perturbation generation phase.** Analytically, we compare the difference between DR-SAM and standard SAM in Tab. 4 from three: (1) the ultimate optimization target, (2) the data sources for acquiring perturbation, and (3) the parameter optimization stage.

(2) **Simple "SAM+OE" cannot enhance detection and generalization simultaneously.** To further justify the validity of aug(·), we simply cooperate auxiliary outliers to obtain perturbation for SAM optimization, termed "SAM+OE." We compare the SAM+OE with standard SAM and proposed DR-SAM in Tab. 5 and 6, which indicates DR-SAM alleviates the issue of training with OE by improving the model's generalization ability (OOD-G accuracy). We provide further discussion in Appendix B.1.

## B  FURTHER EXPERIMENTS

In this section, we conduct experiments to show the effect of data augmentation in DR-SAM. We also provide ablation studies on data augmentation to show the capability for OOD-G of DR-SAM is derived from data augmentation.

### B.1  THE EFFECT OF DATA AUGMENTATION

We first show that model training with auxiliary outliers would affect the ID performance, even with SAM. We then demonstrate that data augmentation could enhance the model's detection and

Table 4: Comparison with SAM

| | SAM | DR-SAM |
|---|---|---|
| Optimization targets | Enhance the model's generalization ability on $D_{ID}$. | Enhanced model's generalization ability on $D_{CS}$ during optimizing with auxiliary outliers to boost the detection ability on $D_{SS}$, which effectively handles the detection-generalization paradox. |
| Acquire perturbation | $D_{ID}^{\text{train}}$ (only pursue low sharpness in $D_{ID}^{\text{train}}$. It does not explicitly involve either covariate-shifted or semantic-shifted samples, and cannot guarantee the model's detection performance under fine-grained datasets like CIFAR-100 or ImageNet-200.) | $aug(D_{ID}^{\text{train}})$ with auxiliary outliers (pursue low sharpness in both $D_{ID}^{\text{train}}$ and $aug(D_{ID}^{\text{train}})$. DR-SAM obtains perturbation using both simulated covariate-shifted samples and auxiliary semantic-shifted data, to create a challenging perturbation aware of the worst case for generalization under covariate shift, while exploring the separation between $D_{ID}$ and $D_{SS}$. |
| Gradient optimization | Perform gradient descent using gradient signal from the perturbed point. This process only guarantees the model's training trajectory for ID classification. | Perform gradient descent using gradient signal from the perturbed point. This process involves samples from $D_{ID}^{\text{train}}$ and auxiliary outliers, allowing us to preserve the benefits of the training trajectory for OOD-D. This process enables us to improve generalization and detection simultaneously in a more harmonious manner. |

Table 5: CIFAR-10 OOD-D and OOD-G performance.

| | FPR@95↓ | AUROC↑ | ID Accuracy↑ | OOD-G Accuracy↑ |
|---|---|---|---|---|
| SAM | 25.15 | 92.08 | 95.69 | 80.69 |
| SAM+OE | 30.95 | 90.94 | 95.27 | 80.24 |
| DR-SAM | 12.36 | 97.04 | 95.13 | 80.50 |

Table 6: CIFAR-100 OOD-D and OOD-G performance.

| | FPR@95↓ | AUROC↑ | ID Accuracy↑ | OOD-G Accuracy↑ |
|---|---|---|---|---|
| SAM | 53.91 | 80.14 | 78.53 | 37.20 |
| SAM+OE | 55.66 | 79.56 | 77.62 | 36.77 |
| DR-SAM | 38.18 | 86.93 | 77.91 | 36.91 |

generalization ability. We conduct experiments on CIFAR-10 and CIFAR-100 with the same setting adopted in Tab. 1 and 2.

As can be seen from Tab. 7 and 8, **model training with auxiliary outliers would sacrifice the model's ID performance but improve its detection capability.** Simply cooperating OE with SAM can improve the model's ID performance, but cannot compete with vanilla SAM. In addition, SAM+OE also fails to achieve competitive performance with vanilla SAM. DR-SAM exceeds SAM+OE in CIFAR-100. This indicates the necessity of the aug(·) for enhancing both detection and generalization performance. In general, despite the slight drawback in ID performance compared to vanilla SAM, DR-SAM effectively resolves the detection-generalization paradox.

## B.2 ABLATION STUDIES ON DATA AUGMENTATION

We conduct experiments on CIFAR-10 and fine-grained dataset ImageNet-200 to demonstrate that the capability for OOD-G of DR-SAM is derived from data augmentation.

As can be seen from Tab. 9 and 10, the model fine-tuned with DR-SAM (w/o aug(·)) would generally decrease the performance of detection and generalization. This indicates that the capability for OOD-G is derived from data augmentation rather than SAM.

Table 7: CIFAR-10 performance comparison.

|        | FPR@95↓ | ID Accuracy↑ |
|--------|---------|--------------|
| MSP    | 39.91   | 95.06        |
| OE     | 14.06   | 95.06        |
| SAM    | 25.15   | 95.69        |
| SAM+OE | 30.95   | 95.27        |
| DR-SAM | 12.36   | 95.13        |

Table 8: CIFAR-100 performance comparison.

|        | FPR@95↓ | ID Accuracy↑ |
|--------|---------|--------------|
| MSP    | 56.68   | 77.26        |
| OE     | 33.20   | 76.51        |
| SAM    | 53.91   | 78.53        |
| SAM+OE | 55.66   | 77.62        |
| DR-SAM | 38.18   | 77.91        |

Table 9: Ablation studies on data augmentation for CIFAR-10.

|                  | FPR@95↓ | OOD-G Accuracy↑ |
|------------------|---------|-----------------|
| MSP              | 39.91   | 95.06           |
| SAM              | 25.15   | 95.69           |
| DR-SAM           | 12.36   | 95.13           |
| DR-SAM (w/o aug($\cdot$)) | 15.46   | 80.16           |

Table 10: Ablation studies on data augmentation for ImageNet-200.

|                  | FPR@95↓ | OOD-G Accuracy↑ |
|------------------|---------|-----------------|
| MSP              | 56.68   | 77.26           |
| SAM              | 53.91   | 78.53           |
| DR-SAM           | 38.18   | 77.91           |
| DR-SAM (w/o aug($\cdot$)) | 45.10   | 44.73           |

## B.3 ADDITIONAL EXPERIMENTS WITH RECENT BASELINES

We provide a comparison of additional baselines, including POEM, NOPS, and NTOM. We additionally conduct experiments with SCONE and WOODS. We conduct experiments with settings aligned with the original submission. As NPOS trains the CNN backbone without the final linear classifier (Zhang et al., 2023b), we can only provide its OOD-D performance and leave the accuracy as N/A. DUL has not yet released the source code, so we will leave the comparison for future work.

We notice that POEM would degenerate the model's classification ability in both in-distribution and covariate-shifted samples, which leads to sub-optimal performance compared to the OE. POEM cannot exceed the OE when using the TIN-597 as auxiliary outliers when training on the CIFAR benchmark, as DUL also has the same report. The WOODS and SCONE, on the other hand, cannot perform well in CIFAR-100 under the traditional OOD-D setting and scarify the OOD-G performance.

## B.4 EXPERIMENTS ON CIFAR DATASET WITH OTHER DATA AUGMENTATION

During training, we choose the brightness as argumentation with the other 17 augmentations for general usage purposes. We do not tune the augmentation based on the performance of the model on the test set. Instead, we still follow the conventional augmentation setup during the training without any specific tuning.

We conducted an experiment on the CIFAR-10C/100C dataset without using brightness augmentation. Specifically, we adopted Gaussian blur and Gaussian noise to augment the test set of the CIFAR datasets and reported the accuracy of the mixed dataset in Tab. 13 and 14. DR-SAM exceeds SAM in terms of OOD-G accuracy on CIFAR-10C/100C without brightness augmentation.

## B.5 FINE-TUNING DR-SAM WITH DIFFERENT DATA AUGMENTATIONS

We employ the AugMix (Hendrycks et al., 2019b) and RandomCrop to train DR-SAM on CIFAR10/100 datasets. Specifically, we follow the experiment settings in Tabs. 1 and 2 for training and evaluation. We also evaluate the model using the Gaussian Blur and Gaussian noise augmented dataset described in Sec. B.4.

For AugMix, we use the PyTorch implementation, set severity as 1, and keep other hyper-parameters as default. We employ RandomCrop with a default setting following the PyTorch implementation.

From the tables shown in Tabs. 15 and 16, we found that Single augmentation might be more effective than the mixing ones, while mixing augmentation would hinder the performance of OOD-D.

From the tables shown above, we found that Single augmentation might be more effective than the mixing ones, while mixing augmentation would hinder the performance of OOD-D.

Table 11: CIFAR-10 OOD-D and OOD-G performance.

|  | FPR@95↓ | AUROC↑ | ID Accuracy↑ | OOD-G Accuracy↑ |
|---|---|---|---|---|
| MSP | 39.91 | 79.24 | 95.06 | 79.24 |
| OE | 14.06 | 96.42 | 95.02 | 78.77 |
| NOPS | 28.05 | 91.85 | N/A | N/A |
| POEM | 33.02 | 88.00 | 85.68 | 66.34 |
| WOODS | 20.36 | 94.39 | 90.43 | 71.47 |
| SCONE | 19.26 | 94.71 | 91.68 | 73.56 |
| DR-SAM | 12.36 | 97.04 | 95.13 | 80.50 |

Table 12: CIFAR-100 OOD-D and OOD-G performance.

|  | FPR@95↓ | AUROC↑ | ID Accuracy↑ | OOD-G Accuracy↑ |
|---|---|---|---|---|
| MSP | 56.68 | 79.02 | 77.26 | 36.59 |
| OE | 36.11 | 87.64 | 76.51 | 36.24 |
| NOPS | 57.22 | 81.06 | N/A | N/A |
| POEM | 40.00 | 82.78 | 68.58 | 32.48 |
| WOODS | 56.15 | 80.32 | 77.01 | 36.48 |
| SCONE | 55.28 | 81.39 | 76.58 | 36.27 |
| DR-SAM | 38.18 | 86.93 | 77.91 | 36.91 |

Table 13: OOD-D and OOD-G performance on CIFAR-10-C.

|  | OOD-G Accuracy↑ |
|---|---|
| MSP | 61.89 |
| OE | 61.32 |
| SAM | 63.17 |
| DR-SAM | 63.56 |

Table 14: OOD-D and OOD-G performance on CIFAR-100-C.

|  | OOD-G Accuracy↑ |
|---|---|
| MSP | 39.50 |
| OE | 36.61 |
| SAM | 40.74 |
| DR-SAM | 39.44 |

- Single augmentation can enhance the model's generalization and detection ability simultaneously. Both Brightness and RandomCrop can enhance the model's detection and generalization performance compared to the OE and MSP baselines. The AugMix, on the other hand, would hinder the model's performance on the in-distribution dataset, and cannot outperform the OE in terms of OOD-D performance.

- Mixed augmentation would cause the model to fail to distinguish semantic-shifted samples from the covariate-shifted ones. We observe that MixOE would also hinder the model's detection ability, which mixes up the $D_{ID}^{train}$ and $D_{SS}^{train}$ to create a smooth transition between two distribution. The observations indicate that these augmentations would weaken the model's ability to distinguish between semantic-shifted samples and covariate-shifted samples. This is likely because the boundary between distributions becomes less defined, making it harder for the model to identify and detect semantic shifts accurately.

For future work, we plan to:

- Conduct extensive experiments on the validation set to uncover the relationships between different augmentation techniques and downstream performance;

- Understanding the impact of augmentation on representation learning to provide more insights into enhancing the model's detection and generalization ability.

In general, the above experiment verifies our claim that "a proper data augmentation can enhance the model's detection and generalization ability simultaneously." We will explore the effect of data augmentation further to provide more insights into how it can enhance the model's detection and generalization ability.

## C  FURTHER DISCUSSION ON THE MODEL'S SHARPENSS

- DR-SAM would have better and steady OOD-D and OOD-G performance when optimized with lower $\rho$. The $\rho$ not only indicates the neighborhood radii but also acts as the hyper-parameter for training the DR-SAM. As shown in Fig. 8(a) and 8(b), a smaller $\rho$ ($\rho$=0.5) is desired to have better detection performance. We also notice that $\rho$ has a relatively weak impact on the model's OOD-G performance within the specific region ($\rho \in [0, 1.5]$). We choose $\rho = 0.5$ to enhance the detection and generalization performance of the model.

Table 15: Performance comparison of DR-SAM fine-tuned with different $aug(\cdot)$ on CIFAR-10.

| | FPR@95↓ | OOD-G Accuracy↑ | OOD-G (Gaussian) Accuracy↑ |
|---|---|---|---|
| MSP | 39.91 | 79.24 | 61.89 |
| OE | 14.06 | 78.77 | 61.32 |
| MixOE | 77.08 | 79.94 | 58.36 |
| DR-SAM (Brightness) | **12.36** | **80.50** | 63.56 |
| DR-SAM (AugMix) | 21.36 | 80.27 | 65.79 |
| DR-SAM (RandomCrop) | 15.46 | 80.16 | **63.25** |

Table 16: Performance comparison of DR-SAM fine-tuned with different $aug(\cdot)$ on CIFAR-100.

| | FPR@95↓ | OOD-G Accuracy↑ | OOD-G (Gaussian) Accuracy↑ |
|---|---|---|---|
| MSP | 56.68 | 36.59 | 39.50 |
| OE | 36.11 | 36.24 | 36.61 |
| MixOE | 57.48 | 36.95 | 36.19 |
| DR-SAM (Brightness) | 38.18 | 36.91 | 39.44 |
| DR-SAM (AugMix) | 49.52 | 35.87 | 37.68 |
| DR-SAM (RandomCrop) | **36.96** | **37.11** | **39.64** |

- The sharpness is only related to the OOD-G performance of the model. Lower sharpness indicates higher OOD-G accuracy (Foret et al., 2021). The sharpness shows less of a relationship with the detection performance of the model.

- A steady low sharpness across range of $\rho$ on $D_{\text{ID}}^{\text{train}}$ might not ensure low sharpness on $D_{\text{ID}}^{\text{test}}$ (Fig. 6 (d)) and $D_{\text{CS}}^{\text{test}}$ (Fig. 7 (d)). Compared to DR-SAM, the pretrained and OE fine-tuned model shows steady low sharpness on $D_{\text{ID}}^{\text{train}}$ (Fig. 5 (d)) but higher sharpness on $D_{\text{CS}}^{\text{test}}$ (Fig. 7 (d)). As a result, their generalization performance cannot exceed the OOD-G methods.

## D DETAILS OF DATASETS

**CIFAR-10** (Krizhevsky & Hinton, 2009) is one of the widely used color image datasets in machine learning, containing real objects in the real world. It has a total of 10 classes which are airplane, automobile, bird, cat, deer, dog, frog, horse, ship, and truck. There are 6000 images for each category, of which 5000 images are used for training and 1000 images are used for testing.

**CIFAR-100** (Krizhevsky & Hinton, 2009) is similar to the CIFAR-10 data in that it has color images of real objects. It contains a total of 100 categories, divided into 20 superclasses. Each class has 600 images, including 500 for training and 100 for testing.

**CIFAR-10/100-C** (Hendrycks & Dietterich, 2019) is obtained by corrupting the original CIFAR test set. It has applied a total of 15 corruptions which are Gaussian Noise, Shot Noise, Impulse Noise, Defocus Blur, Frosted Glass Blur, Motion Blur, Zoom Blur, Snow, Frost, Fog, Brightness, Contrast, Elastic, Pixelate, JPEG.

**Place365** (Zhou et al., 2018) is a scene recognition dataset with a total of 434 scene categories. There are two versions of the dataset which are Places365-Standard and Places365-Challenge-2016. The Place365 has a total of 10 million images, with between 5,000 and 30,000 training images per class.

**MNIST** (Deng, 2012) is a dataset of handwritten number images with 10 classes, each representing a number between 0 and 9. The MNIST dataset has a total of 70,000, 28×28 greyscale images, of which 60,000 are in the training set and 10,000 are in the test set.

**SVHN** (Netzer et al., 2011) is a image datasets with real-world numbers. The numbers in it are captured from various scenes, such as door numbers and historical buildings. It divided into 10 categories and each number in the image is belonged to one class. It contains 73257 digits for training and 26032 digits for testing.

**Texture** (Cimpoi et al., 2014) a real world surface texture dataset. The images are collected from wood, blankets, cloth, leather, etc. There are 64 categories and a total of 8674 images. The Texture is mainly used to evaluate the capabilities of the model or as a pre-training dataset.

**TinyImageNet** (Le & Yang, 2015) contains 100000, 64×64 coloured images with 200 classes. Each class has 500 training images, 50 validation images, and 50 test images.

**TIN-597** (Zhang et al., 2023b) obtained from ImageNet-1K that is not overlapped with TIN dataset. It has 597 classes and is cleared of CIFAR-10/100 related categories.

**ImageNet-200** is the subset from the ImageNet-1K that has same 200 classes as ImageNet-R. In comparison to ImageNet-1K, it contains identical OOD datasets.

**SSB-hard** (Vaze et al., 2021) is the hard split of SSB dataset which has 980 classes and contains 49K images. It used to explore semantic shift tasks and obtained based on fine-grained datasets.

| Method | CIFAR100 | | TIN | | MINIST | | SVHN | | Textures | | Places365 | |
|---|---|---|---|---|---|---|---|---|---|---|---|---|
| | FPR95↓ | AUROC↑ | FPR95↓ | AUROC↑ | FPR95↓ | AUROC↑ | FPR95↓ | AUROC↑ | FPR95↓ | AUROC↑ | FPR95↓ | AUROC↑ |
| Post-hoc | | | | | | | | | | | | |
| MSP | $53.09_{\pm4.89}$ | $87.19_{\pm0.33}$ | $43.18_{\pm3.20}$ | $88.89_{\pm0.20}$ | $23.64_{\pm5.82}$ | $92.63_{\pm1.57}$ | $25.43_{\pm1.57}$ | $91.56_{\pm0.38}$ | $35.20_{\pm4.55}$ | $89.89_{\pm0.71}$ | $42.47_{\pm3.90}$ | $88.92_{\pm0.47}$ |
| RMDS | $43.88_{\pm3.48}$ | $88.83_{\pm0.35}$ | $33.68_{\pm1.67}$ | $90.83_{\pm0.27}$ | $21.49_{\pm2.31}$ | $93.22_{\pm0.80}$ | $23.03_{\pm1.51}$ | $91.95_{\pm0.25}$ | $25.31_{\pm0.56}$ | $92.23_{\pm0.23}$ | $31.18_{\pm0.32}$ | $91.51_{\pm0.11}$ |
| EBO | $66.58_{\pm4.48}$ | $86.36_{\pm0.58}$ | $56.68_{\pm5.36}$ | $88.85_{\pm0.36}$ | $24.98_{\pm12.92}$ | $94.32_{\pm2.53}$ | $34.21_{\pm5.55}$ | $91.98_{\pm0.91}$ | $52.00_{\pm6.23}$ | $89.47_{\pm0.69}$ | $54.68_{\pm6.62}$ | $89.25_{\pm0.78}$ |
| VIM | $49.32_{\pm3.10}$ | $87.75_{\pm0.28}$ | $40.20_{\pm1.38}$ | $89.71_{\pm0.32}$ | $18.36_{\pm1.42}$ | $94.75_{\pm0.38}$ | $18.89_{\pm0.58}$ | $94.59_{\pm0.47}$ | $21.18_{\pm1.62}$ | $95.14_{\pm0.32}$ | $41.47_{\pm2.22}$ | $89.49_{\pm0.38}$ |
| KNN | $37.63_{\pm0.29}$ | $89.73_{\pm0.14}$ | $30.20_{\pm0.71}$ | $91.62_{\pm0.27}$ | $20.04_{\pm1.35}$ | $94.26_{\pm0.23}$ | $22.39_{\pm1.39}$ | $92.77_{\pm0.30}$ | $24.06_{\pm0.46}$ | $93.16_{\pm0.23}$ | $30.35_{\pm0.66}$ | $91.77_{\pm0.23}$ |
| Training Methods (w/o Outlier Data) | | | | | | | | | | | | |
| ConfBranch | $39.03_{\pm0.39}$ | $86.67_{\pm0.39}$ | $34.90_{\pm0.65}$ | $88.06_{\pm0.28}$ | $19.81_{\pm1.41}$ | $92.89_{\pm1.20}$ | $25.09_{\pm1.18}$ | $89.49_{\pm0.35}$ | $42.42_{\pm1.00}$ | $85.30_{\pm0.15}$ | $35.49_{\pm1.42}$ | $87.43_{\pm0.84}$ |
| G-ODIN | $64.92_{\pm5.16}$ | $83.51_{\pm1.52}$ | $61.73_{\pm5.74}$ | $85.18_{\pm1.25}$ | $24.89_{\pm4.87}$ | $94.85_{\pm0.95}$ | $57.19_{\pm10.20}$ | $85.41_{\pm2.56}$ | $67.43_{\pm7.80}$ | $85.31_{\pm1.66}$ | $48.66_{\pm3.23}$ | $88.61_{\pm0.46}$ |
| LogitNorm | $32.58_{\pm0.71}$ | $91.18_{\pm0.23}$ | $24.06_{\pm1.07}$ | $93.62_{\pm0.26}$ | $2.55_{\pm1.28}$ | $99.45_{\pm0.31}$ | $11.11_{\pm1.01}$ | $97.05_{\pm0.22}$ | $24.18_{\pm3.73}$ | $93.64_{\pm0.68}$ | $22.82_{\pm0.56}$ | $94.05_{\pm0.11}$ |
| Training Methods (w/ Outlier Data) | | | | | | | | | | | | |
| OE | $32.74_{\pm3.01}$ | $91.56_{\pm0.79}$ | $2.09_{\pm1.60}$ | $99.45_{\pm0.29}$ | $17.86_{\pm0.50}$ | $94.52_{\pm0.36}$ | $0.42_{\pm0.25}$ | $99.84_{\pm0.08}$ | $10.86_{\pm1.31}$ | $98.00_{\pm0.33}$ | $13.67_{\pm1.40}$ | $96.96_{\pm0.48}$ |
| MCD | $41.10_{\pm0.44}$ | $87.19_{\pm0.06}$ | $35.86_{\pm0.33}$ | $88.68_{\pm0.09}$ | $24.45_{\pm2.12}$ | $92.42_{\pm0.48}$ | $34.94_{\pm3.58}$ | $88.02_{\pm1.08}$ | $37.08_{\pm2.16}$ | $88.04_{\pm0.26}$ | $36.34_{\pm0.89}$ | $88.60_{\pm0.18}$ |
| MixOE | $90.24_{\pm4.98}$ | $83.54_{\pm1.88}$ | $84.82_{\pm7.54}$ | $86.84_{\pm1.68}$ | $65.27_{\pm10.19}$ | $90.41_{\pm0.70}$ | $32.10_{\pm24.71}$ | $93.23_{\pm1.80}$ | $84.99_{\pm14.40}$ | $85.54_{\pm3.70}$ | $84.19_{\pm7.24}$ | $87.25_{\pm1.47}$ |
| Ours | | | | | | | | | | | | |
| DR-SAM | $31.01_{\pm1.47}$ | $92.23_{\pm0.21}$ | $6.60_{\pm1.47x}$ | $98.59_{\pm0.32}$ | $4.59_{\pm3.34}$ | $98.80_{\pm0.92}$ | $1.59_{\pm0.51}$ | $99.55_{\pm0.16}$ | $6.49_{\pm1.68}$ | $98.73_{\pm0.35}$ | $10.95_{\pm1.21}$ | $97.58_{\pm0.40}$ |

Table 17: CIFAR-10 full results

| Method | CIFAR10 | | TIN | | MINIST | | SVHN | | Textures | | Places365 | |
|---|---|---|---|---|---|---|---|---|---|---|---|---|
| | FPR95↓ | AUROC↑ | FPR95↓ | AUROC↑ | FPR95↓ | AUROC↑ | FPR95↓ | AUROC↑ | FPR95↓ | AUROC↑ | FPR95↓ | AUROC↑ |
| Post-hoc | | | | | | | | | | | | |
| MSP | $58.90_{\pm0.93}$ | $78.47_{\pm0.07}$ | $50.78_{\pm0.57}$ | $81.96_{\pm0.20}$ | $57.24_{\pm4.67}$ | $76.08_{\pm1.86}$ | $58.42_{\pm2.62}$ | $78.68_{\pm0.95}$ | $61.78_{\pm1.30}$ | $77.32_{\pm0.71}$ | $56.64_{\pm0.87}$ | $79.22_{\pm0.29}$ |
| RMDS | $61.37_{\pm0.23}$ | $77.75_{\pm0.19}$ | $49.50_{\pm0.64}$ | $82.58_{\pm0.02}$ | $52.04_{\pm6.27}$ | $79.74_{\pm2.49}$ | $51.06_{\pm3.56}$ | $85.10_{\pm1.06}$ | $53.95_{\pm0.99}$ | $83.65_{\pm0.52}$ | $53.56_{\pm0.37}$ | $83.40_{\pm0.46}$ |
| EBO | $59.18_{\pm0.75}$ | $79.05_{\pm0.10}$ | $52.35_{\pm0.58}$ | $82.58_{\pm0.08}$ | $52.61_{\pm3.84}$ | $79.18_{\pm1.36}$ | $53.19_{\pm3.25}$ | $82.28_{\pm1.79}$ | $62.39_{\pm2.07}$ | $78.35_{\pm0.84}$ | $57.70_{\pm0.85}$ | $79.50_{\pm0.23}$ |
| VIM | $70.59_{\pm0.42}$ | $72.21_{\pm0.41}$ | $54.63_{\pm0.41}$ | $77.87_{\pm0.13}$ | $48.31_{\pm1.06}$ | $81.88_{\pm1.02}$ | $46.29_{\pm5.47}$ | $82.91_{\pm3.77}$ | $46.84_{\pm2.26}$ | $85.91_{\pm0.78}$ | $61.57_{\pm0.74}$ | $75.86_{\pm0.37}$ |
| KNN | $72.81_{\pm0.45}$ | $77.02_{\pm0.25}$ | $49.55_{\pm0.54}$ | $83.31_{\pm0.16}$ | $48.58_{\pm4.68}$ | $82.36_{\pm1.52}$ | $51.48_{\pm3.21}$ | $84.27_{\pm1.10}$ | $53.62_{\pm2.38}$ | $83.66_{\pm0.84}$ | $60.76_{\pm0.90}$ | $79.42_{\pm0.47}$ |
| Training Methods (w/o Outlier Data) | | | | | | | | | | | | |
| ConfBranch | $83.84_{\pm0.48}$ | $63.58_{\pm0.19}$ | $72.23_{\pm0.69}$ | $71.03_{\pm0.06}$ | $39.36_{\pm9.97}$ | $86.70_{\pm5.83}$ | $84.57_{\pm2.39}$ | $51.46_{\pm2.68}$ | $96.67_{\pm0.84}$ | $50.97_{\pm0.28}$ | $76.77_{\pm0.91}$ | $66.80_{\pm0.42}$ |
| G-ODIN | $74.00_{\pm7.25}$ | $71.59_{\pm3.20}$ | $62.15_{\pm4.82}$ | $77.84_{\pm2.66}$ | $49.33_{\pm4.60}$ | $82.66_{\pm2.72}$ | $54.79_{\pm3.93}$ | $76.45_{\pm2.04}$ | $54.52_{\pm2.85}$ | $80.88_{\pm2.01}$ | $67.05_{\pm4.98}$ | $75.22_{\pm3.68}$ |
| LogitNorm | $67.91_{\pm0.43}$ | $76.30_{\pm0.32}$ | $50.08_{\pm0.50}$ | $83.07_{\pm0.31}$ | $27.77_{\pm2.56}$ | $92.25_{\pm0.88}$ | $46.71_{\pm6.78}$ | $82.18_{\pm4.22}$ | $63.19_{\pm2.24}$ | $77.56_{\pm1.50}$ | $56.76_{\pm0.62}$ | $79.63_{\pm0.50}$ |
| Training Methods (w/ Outlier Data) | | | | | | | | | | | | |
| OE | $65.47_{\pm0.90}$ | $74.69_{\pm0.74}$ | $0.94_{\pm0.63}$ | $99.63_{\pm0.14}$ | $44.88_{\pm4.53}$ | $86.33_{\pm3.51}$ | $2.06_{\pm0.88}$ | $99.29_{\pm0.22}$ | $52.17_{\pm4.33}$ | $86.09_{\pm2.01}$ | $57.01_{\pm2.23}$ | $80.76_{\pm1.27}$ |
| MCD | $62.63_{\pm0.20}$ | $75.95_{\pm0.16}$ | $54.72_{\pm0.42}$ | $79.53_{\pm0.21}$ | $64.69_{\pm3.01}$ | $73.02_{\pm1.82}$ | $55.57_{\pm3.47}$ | $79.43_{\pm1.09}$ | $68.38_{\pm0.49}$ | $73.94_{\pm0.15}$ | $59.42_{\pm0.12}$ | $77.24_{\pm0.17}$ |
| MixOE | $62.33_{\pm2.15}$ | $78.42_{\pm0.64}$ | $51.44_{\pm1.98}$ | $82.46_{\pm0.75}$ | $60.06_{\pm7.82}$ | $73.65_{\pm4.95}$ | $50.21_{\pm7.14}$ | $82.75_{\pm2.55}$ | $63.50_{\pm2.40}$ | $78.76_{\pm0.50}$ | $58.47_{\pm1.53}$ | $79.64_{\pm0.52}$ |
| Ours | | | | | | | | | | | | |
| DR-SAM | $59.05_{\pm0.42}$ | $78.39_{\pm0.10}$ | $0.11_{\pm0.05}$ | $99.88_{\pm0.04}$ | $54.32_{\pm5.65}$ | $77.26_{\pm3.16}$ | $29.17_{\pm3.80}$ | $94.73_{\pm0.95}$ | $49.81_{\pm0.97}$ | $85.42_{\pm0.17}$ | $53.76_{\pm0.51}$ | $81.52_{\pm0.17}$ |

Table 18: CIFAR-100 full results

**NINCO** (Bitterwolf et al., 2023) contains 5879 noise-free images.

**iNaturalist** (Van Horn et al., 2018) has 13 super-classes which included Plantae, Insecta, Aves, Mammalia and so on. It has 675170 images for training and validation.

**OpenImage-O** (Wang et al., 2022) contains images are selected one by one from the OpenImage-V3 test set.

**ImageNet-R** (Hendrycks et al., 2021) has 200 classed of ImageNet dataset containing 30000 images.

# E  FULL EXPERIMENT RESULTS

In this part, we report the full training performance for CIFAR-10, CIFAR-100, and ImageNet-200 in Tab. 17, Tab. 18, and Tab. 3.

# F  CONFIGURATION

We follow the benchmark setting introduced in OpenOOD (Yang et al., 2022).

**Model fine-tuning configurations.** For both CIFAR-10 and CIFAR-100, we run for 100 epochs with an initial learning rate of 0.01 and the ReduceLROnPlateau learning rate scheduler with the patience parameter of 5. For ImageNet-200, we employ 0.001 fine-tuning learning rate. The batch size is 128 for $D_{\text{ID}}$ and 256 for $D_{\text{OOD}}^{\text{train}}$. We adopt SGD optimizer with Nesterov momentum (Sutskever et al., 2013) that is set as 0.9, and the weight decay is set as $5e^-4$. We ran our experiment under three seeds, reporting their mean and standard deviation.

**DR-SAM configurations.** For CIFAR-10, we employ $\lambda$ and $\rho$ as 0.5, and using brightness as $\text{aug}(\cdot)$. For CIFAR-100, we employ we employ $\lambda$ and $\rho$ as 0.2, and using brightness as $\text{aug}(\cdot)$. For ImageNet-200, we $\lambda$ and $\rho$ as 0.5.

| Method | SBHARD | | NINCO | | iNATURALIST | | Textures | | OPENIMAGE_O | |
|---|---|---|---|---|---|---|---|---|---|---|
| | FPR95↓ | AUROC↑ | FPR95↓ | AUROC↑ | FPR95↓ | AUROC↑ | FPR95↓ | AUROC↑ | FPR95↓ | AUROC↑ |
| Post-hoc | | | | | | | | | | |
| MSP | 66.09±0.05 | 80.32±0.04 | 43.76±0.72 | 86.27±0.11 | 26.53±0.69 | 92.77±0.26 | 44.43±0.73 | 88.37±0.13 | 35.22±0.26 | 89.24±0.01 |
| RMDS | 66.02±0.33 | 80.16±0.25 | 42.17±1.12 | 84.92±0.27 | 24.74±0.91 | 90.62±0.47 | 37.93±1.15 | 86.76±0.40 | 34.83±0.40 | 86.78±0.22 |
| EBO | 69.97±0.26 | 79.72±0.03 | 50.66±0.94 | 85.10±0.12 | 26.41±2.29 | 92.52±0.51 | 41.31±1.84 | 90.80±0.16 | 36.81±1.15 | 89.20±0.26 |
| VIM | 71.51±0.47 | 73.90±0.31 | 47.21±1.15 | 83.29±0.18 | 27.37±0.39 | 90.97±0.36 | 20.39±0.18 | 94.61±0.11 | 33.91±0.72 | 88.20±0.18 |
| KNN | 73.89±0.27 | 76.88±0.24 | 46.74±0.78 | 86.05±0.12 | 24.43±1.10 | 93.97±0.36 | 24.53±0.21 | 95.30±0.02 | 32.94±1.11 | 90.17±0.32 |
| Training Methods (w/o Outlier Data) | | | | | | | | | | |
| ConfBranch | 72.21±0.11 | 75.06±0.27 | 50.91±0.83 | 82.94±0.12 | 23.80±1.15 | 93.40±0.26 | 39.91±0.53 | 90.03±0.15 | 36.97±0.38 | 88.48±0.18 |
| G-ODIN | 77.66±0.32 | 73.33±0.22 | 60.19±0.95 | 81.92±0.12 | 26.86±1.24 | 92.64±0.22 | 26.81±1.74 | 93.96±0.09 | 35.41±0.44 | 90.15±0.13 |
| LogitNorm | 67.13±0.55 | 78.45±0.18 | 47.60±0.62 | 86.01±0.16 | 16.21±1.22 | 96.10±0.34 | 32.02±1.25 | 92.00±0.19 | 31.61±1.06 | 90.66±0.45 |
| Training Methods (w/ Outlier Data) | | | | | | | | | | |
| OE | 63.44±1.67 | 82.15±0.36 | 41.44±0.34 | 86.93±0.21 | 29.85±0.28 | 89.22±0.30 | 44.71±2.72 | 86.56±0.79 | 35.69±1.33 | 87.86±0.44 |
| MCD | 68.03±0.11 | 76.59±0.06 | 51.17±1.18 | 81.52±0.17 | 35.39±0.40 | 89.18±0.23 | 51.77±1.41 | 84.84±0.26 | 46.27±0.54 | 84.23±0.16 |
| MixOE | 71.49±0.20 | 79.53±0.22 | 49.51±2.68 | 84.64±0.64 | 32.43±2.13 | 89.81±0.61 | 54.00±2.10 | 86.02±0.56 | 40.64±2.02 | 87.15±0.55 |
| Ours | | | | | | | | | | |
| DR-SAM | 63.08±0.45 | 82.88±0.23 | 41.39±0.79 | 87.33±0.21 | 26.77±0.30 | 90.22±0.09 | 41.20±1.16 | 88.08±0.37 | 34.07±0.72 | 88.75±0.17 |

Table 19: ImageNet-200 full results

**Visualization configurations.** For Fig. 3 and Fig. 9, we employ CIFAR-10 as ID, CIFAR-10-C as CSID, MNIST as SSID. The same dataset setting for the landscapes is shown in Fig. 10 (a-b). For Fig. 10 (c-d), we employ CIFAR-100 as ID, CIFAR-100-C as CSID, and MNIST as SSID.

# G RELATED WORK

## G.1 OUT-OF-DISTRIBUTION GENERALIZATION

Empirical Risk Minimization (Vapnik, 1998) methods are insufficient for generalizing novel test distributions because they rely on spurious correlations that only exist in the training data.

Several methods based on representation learning are proposed to generalize the model to new data distribution (Li et al., 2018b; Huang et al., 2020; Xu & Jaakkola, 2021; Lu et al., 2022; Sun & Saenko, 2016; Zhang et al., 2021; Kim et al., 2021). The representative Invariant Risk Minimization (IRM) (Arjovsky et al., 2019) identifies and removes spurious correlations by learning invariant representations of the data. GroupDRO (Sagawa et al., 2020) trains the model to perform well not only on the average data distribution but also on a set of "nearby" distributions defined by a given uncertainty set. VRE-x (Krueger et al., 2021) considers optimizing the affine combinations of training risks instead of the convex combinations of the training risks adopted in GroupDRO.

## G.2 OUT-OF-DISTRIBUTION DETECTION

To identify the out-of-distribution samples from in-distribution ones, training-free post-hoc OOD-D methods mainly modify the model's softmax prediction probability to enlarge the gap between in-distribution and out-of-distribution samples (Hendrycks & Gimpel, 2016; Liang et al., 2017; Liu et al., 2020). Another line is to focus on modifying the model's representation to identify the out-of-distribution samples (Zhu et al., 2022; Bitterwolf et al., 2020; Huang et al., 2022; Ming et al., 2022b; Djurisic et al., 2022). In contrast, some researchers focus on modifying the model's activation value to identify the out-of-distribution samples, including ReAct Sun et al. (2021), ASH Djurisic et al. (2022), CONFBRANCH DeVries & Taylor (2018), T2FNorm Regmi et al. (2023), Logitnorm Wei et al. (2022), and Tian et al. (2021).

To further enhance the model's detection ability, Outlier Expose (OE) (Hendrycks et al., 2018) training the model with auxiliary outliers to allow the model to be aware of the semantic-shifted samples. Based on OE, DOE (Wang et al., 2023) uses the implicit transformed data produced by model perturbation to expand distributions for training. MixOE (Zhang et al., 2023a) solves this problem by adopting Mixup or Cutup to combine ID data and surrogate data, which generates a new dataset for training. DivOE (Zhu et al., 2023) provides an adversarial training approach to generate novel and challenging outliers to enhance the detection performance. POEM Ming et al. (2022a) uses an auxiliary outlier dataset to update the posterior distribution's decision boundary between OOD

and ID data. By jointly modeling the ID and OOD data, the UDG framework Yang et al. (2021) can enrich the semantic knowledge of the model by exploiting unlabeled data in an unsupervised manner

However, the OE approaches require access to the auxiliary outliers, which might limit their application. G-ODIN Hsu et al. (2020) proposes two corresponding strategies to improve the performance of OOD detection under the setting that no additional OOD data is used for fine-tuning. Hendrycks et al. (2019a) found that without using a large model or additional data, the self-supervised models obtained are more robust regarding adversarial robustness, label corruption, common input corruptions, and out-of-distribution detection. Based on this phenomenon, ROTPRED Hendrycks et al. (2019a), which is a self-supervised model, learns representations that favor downstream tasks such as OOD detection by predicting the angle of rotation. VOS Du et al. (2022) sampling outliers from the low-likelihood region of the ID data and training the model with the ID. VOS synthetic OOD data to obtain a decision boundary that improves the model's OOD-D performance.

Some existing OOD detection models are trained based on small, low-resolution datasets, such as CIFAR-10, and the models cannot be transferred to large-scale settings. MOS Huang & Li (2021) forms several classes into a new group by taxonomy feature clustering or random grouping to simplify OOD and ID data's decision boundary. The model detects OOD data based on the total confidence values of the input data maps to other classes in all new groups.

The limitations of current OOD detection benchmarks have overcome some challenges that may identify data with the same semantics but different sources as OOD. Yang et al. (2021) proposes a novel Semantically Coherent Out-of-Distribution Detection (SC-OOD) benchmark that evaluates the ability of models to detect OOD samples that are semantically coherent with the ID samples.

### G.3    METHODS CONSIDER OOD-D AND OOD-G.

The model would encounter different distributional shifts when deployed in the wild. The pioneering work (Bai et al., 2023) proposes to enhance the model's OOD-D and OOD-G ability when training the model with wild data. [1] proposes SCONE to handle the wild data, assuming that semantic shifts would be encountered less frequently. SCONE forces the model to lower the wild samples' energy while enforcing a sufficient margin between the $D_{\text{ID}}^{\text{train}}$ and a pre-defined energy threshold. Since the model would allocate ID samples with lower energy scores than the semantic-shifted ones, the former objective allows the model to detect the semantic-shifted samples. In contrast, the latter allows the model to predict the covariate-shifted samples correctly.

Averly & Chao (2023) proposes an OOD-D evaluation framework to detect and reject the misclassified covariate-shifted samples while accepting the correctly classified ones. The proposed evaluation framework mainly identifies the sample model cannot predict correctly regardless of their distribution shift types. Averly & Chao (2023) does not explicitly argue the detection-generalization dilemma or propose a new algorithm to enhance both detection and generalization ability.

The concurrent work (Zhang et al., 2024) shares the same setting as DR-SAM but develops a different method to enhance the model's generalization and detection. Zhang et al. (2024) propose a novel and theoretical guarantee optimization framework, Decoupled Uncertainty Learning (DUL), to enhance the model's OOD-D ability while maintaining the original OOD-G capability. DUL incorporates distributional uncertainty in the Bayesian framework to bridge the detection and generalization learning target. Specifically, DUL encourages the exposed outlier to have high uncertainty while maintaining a non-increased overall uncertainty to ensure generalization capability.

### G.4    ROBUST FINETUNING

Pham et al. (2023) shows that the model fine-tuned on a dataset with shifted distribution would have lower performance than that of the original zero-shot model. This means that the fine-tuning of the model sacrifices its robustness. WiSE-FT (Wortsman et al., 2022) finds that small variations in the hyperparameters lead to variable model accuracies and that aggressive fine-tuning may lead to reduced accuracy in the distribution shift target dataset. Chen et al. (2020) introduces adversarial training with self-supervised learning to pre-train and fine-tune the model. AFT Jeddi et al. (2020) uses 'Slow Start, Fast Decay' fine-tuning to improve the robustness of the model by controlling the learning rate during the fine-tuning phase, which uses adversarial perturbations.

