# OpenReview forum: "Breaking the Detection-Generalization Paradox on Out-Of-Distribution Data"
_ICLR.cc/2025/Conference — Submitted to ICLR 2025_

### Official Review · Reviewer_5Bu7 · 2024-10-18

**Soundness:** 3
**Presentation:** 3
**Contribution:** 1
**Rating:** 6
**Confidence:** 4

**Summary:**

The paper focuses on the relationship between two widely studied problems, i.e., OOD generalization and OOD detection. The authors conducted an empirical study and found that these two tasks conflicted with many previous OOD detection methods. To address this issue, a novel optimization framework named DR-SAM is proposed to balance OOD generalization and detection performance. The proposed method obtains the optimal model parameters by pursuing not only a clear decision boundary between clean ID data and semantic OOD data but also simulated covariant-shifted data and semantic OOD data. And thus better overall performance can be expected. Experiments on commonly used benchmarks can support the proposed method.

**Strengths:**

## Strength

- Interesting topic. The investigated problem is realistic, practical, and important. Combining these two tasks is necessary and critical.
- Clear writing and good organization. The logic of the most part in this paper is smooth which makes it easy to follow. I enjoy the clear writing.

**Weaknesses:**

## Weakness

Major concerns:

- Vague contributions. The authors claim that the main contribution of this paper is to identify the detection-generalization paradox. However, as far as I could tell, the trade-off/conflict relationship has been pointed out by several previous works [1] [2] [3]. Thus such a claim should be either toned down or a clear explanation about the difference from previous work should be provided.
- Unclear motivation. In lines 266-269, the authors claim that the ideal model should yield low sharpness on both ID and covariate-shifted data. They claim that this cannot be adopted OOD-G method. The logic here is hard for me to follow. Why solely using OOD-G method can not ensure low sharpness for ID and covariate-shifted data? I guess this is a typo. Is the covariate-shifted data here should be replaced with semantic-shifted data?
- Lock of essential comparison. Although the experiments in Section 5 encompass a few representative works in OOD detection and generalization, some of the most related works are missed. Several recent methods also jointly consider OOD detection and generalization [1] [3] [4]. Thus, the comparison in this current version is biased and incomplete. Besides, I also feel that some SOTA OOD detection methods are also missed. For example, POEM [5], NPOS[6], and NTOM[7]. As far as I can tell, POEM substantially surpasses OE, MCD, and MixOE in terms of OOD detection on CIFAR benchmarks. SCONE, WOODS, and DUL which jointly pursue OOD detection and generalization can achieve much better overall performance compared to the baselines in Table 1 2 and 3. The reviewer suggests comparing the proposed method with these methods.
- Experimental settings. The authors claim that they deploy brightness as data augmentation (Appendix C). I have concerns about whether using brightness augmentation during training can result in information leakage from the test covariate-shifted distribution. Since all the other corruptions in CIFAR10/100-C or ImageNet-C may also alter the brightness of images. As far as I could tell, in standard OOD generalization settings, the test-time covariate-shifted distribution should be kept unknown during training. Besides, the authors seem to tune the augmentation and make such a choice as they said in lines 466-468. Thus, the dependence on manually selected augmentation is a notable limitation that makes the OOD generalization problem more like domain adaption or even a trivial problem.

Minor concerns:

- Similar to other training-required OOD detection methods, the proposed method also needs to access semantic-shifted data during training. This limitation widely exists in many previous OOD detection methods, but still worth noting here.

- In Figure 3(d), why there is no blue area (ID data) in the figure?
- I am unsure whether the formulation of OOD generalization in Eq. 3 is correct.  $D_{CS}$ is mentioned by the text above Eq.3. However, there is no such notation in the following equation. The formulation should be revised carefully and a proper citation should be provided here.
- The authors post a visualization of the loss landscape in Figure 10. However, such a visualization contains limited information. The reviewer suggests comparing with SAM, OE, and the original ERM.

Overall, the quality of this paper in its current version does not meet the expectations of ICLR. However, I may adjust my score according to opinions from other reviewers if a strong argument is provided.

[1] Feed two birds with one scone: Exploiting wild data for both out-of-distribution generalization and detection. ICML'23

[2] Unified out-of-distribution detection: A model-specific perspective. CVPR'23

[3] The Best of Both Worlds: On the Dilemma of Out-of-distribution Detection. NIPS'24

[4] Training ood detectors in their natural habitats. ICML'22

[5] Poem: Out-of-distribution detection with posterior sampling. ICML'22

[6] Non-parametric outlier synthesis. ICLR'23

[7] Atom: Robustifying out-of-distribution detection using outlier mining. ECML PKDD'21

**Questions:**

See above.

---

> ### Author Response · Authors · 2024-11-22
> **Response to Reviewer 5Bu7 (1/3)**
>
> We thank the reviewer 5Bu7 for the valuable feedback. We addressed all the comments. Please kindly find the detailed responses below. Any further comments and discussions are welcomed!
>
> **Q1:** About the contribution.
>
> > Vague contributions. The authors claim that the main contribution of this paper is to identify the detection-generalization paradox. However, as far as I could tell, the trade-off/conflict relationship has been pointed out by several previous works [1] [2] [3]. Thus such a claim should be either toned down or a clear explanation about the difference from previous work should be provided.
>
> **Reply:** Thanks for this valuable feedback. **Please note that we identify the paradox with $D_{\text{ID}}^{\text{train}}$ and $D_{\text{SS}}^{\text{train}}$ instead of the wild data in [1,2]. In addition, we develop the model under the framework of SAM, which is orthogonal to the Bayesian framework in [3].**
>
> We provide a comprehensive comparison of the submission and the previous works.
>
> |             |                          SCONE [1]                           | [2]                                                          | DUL [3]                                                      | DR-SAM (ours)                                                |
> | ----------- | :----------------------------------------------------------: | ------------------------------------------------------------ | ------------------------------------------------------------ | ------------------------------------------------------------ |
> | Setting     | **Models are fine-tuned with samples from the wild.** The wild data contain samples from mixed types of distributions, including covariate-shift and semantic-shift data. | **[2] focuses on detecting and rejecting samples beyond the model's classification capability**, which aims to evaluate the existing model's robustness with samples from mixing distribution types. Conversely, we focus on enhancing the model's generalization and detection ability simultaneously. | Training distribution contains **only labeled ID samples and unlabeled auxiliary outliers**. | Training distribution contains **only labeled ID samples and unlabeled auxiliary outliers**. |
> | Target      | Training the model to handle samples **from mixed types of distribution**, termed wild data. | Detecting and rejecting samples **beyond the model's classification capability. ** | **Maintaining** model's OOD-G capability while **enhancing** its detection performance. | **Enhance** model's detection and generlization  performance  **simultaneously. ** |
> | Methodology | **Forcing in-distribution samples to have low energy**, with the assumption that semantic shifts would be encountered less frequently. | **NONE** (as this paper focuses on evaluation.)              | **Decoupling** the **uncertainty**  under the **Bayesian framework.** | **Minimize** the model's **loss sharpness** under the **framework of SAM.** |
>
> We have updated the above discussion in the main text in our latest submission.
>
> **Q2:** About the motivation.
>
> > Unclear motivation. In lines 266-269, the authors claim that the ideal model should yield low sharpness on both ID and covariate-shifted data. They claim that this cannot be adopted OOD-G method. The logic here is hard for me to follow. Why solely using OOD-G method can not ensure low sharpness for ID and covariate-shifted data? I guess this is a typo. Is the covariate-shifted data here should be replaced with semantic-shifted data?
>
> **Reply:** Thanks for this valuable feedback. **Basically, the OOD-G methods cannot ensure lower sharpness as they focus on loss consistency across domains.**
>
> We would like to (1) clarify the reason for OOD-G method could not ensure low sharpness and (2) explain our motivation further.
>
> - **OOD-G methods, such as GroupDRO and VRE-x, focus on loss consistency across domains rather than lowering the sharpness.** In addition, the differences in data amounts and task difficulty of different domains in OOD-G would result in biased sharpness estimation towards specific domains, which may lead to a suboptimal sharp minima with a similar loss scale [1]. As OOD-G methods do not explicitly consider the sharpness during optimization, training with OOD-G cannot guarantee low sharpness.

---

> ### Author Response · Authors · 2024-11-22
> **Response to Reviewer 5Bu7 (2/3)**
>
> - **An ideal model should have low sharpness on both in-distribution and covariate-shift data.** We observes that OOD-G fine-tuned model have **higher sharpness in $D_{\text{ID}}^{\text{train}}$ and $D_{\text{ID}}^{\text{test}}$ but lower in $D_{\text{CS}}^{\text{test}}$** compare to pre-trained model. Subsequently, the OOD-G fine-tuned model **generalize well in $D_{\text{CS}}^{\text{test}}$ but fail to achieve competitive result in $D_{\text{ID}}^{\text{train}}$ and $D_{\text{ID}}^{\text{test}}$**, as shown in Tab. 1 and 2. Since a lower sharpness is desired for the model to generalize well on the specific distribution (lines 256-257 in the original submission), such an observation leads to the claim that: "an ideal model should have low sharpness on both in-distribution and covariate-shift data."
>
> In general, training with OOD-G can not ensure low sharpness for $D_{\text{ID}}$ and $D_{\text{test}}$ based on our observation and discussion in [1].
>
> [1] Domain-Inspired Sharpness-Aware Minimization Under Domain Shifts. In ICLR, 2024.
>
> **Q3:** Comparing additional methods.
>
> > Lock of essential comparison. Although the experiments in Section 5 encompass a few representative works in OOD detection and generalization, some of the most related works are missed. Several recent methods also jointly consider OOD detection and generalization [1] [3] [4]. Thus, the comparison in this current version is biased and incomplete. Besides, I also feel that some SOTA OOD detection methods are also missed. For example, POEM [5], NPOS[6], and NTOM[7]. As far as I can tell, POEM substantially surpasses OE, MCD, and MixOE in terms of OOD detection on CIFAR benchmarks. SCONE, WOODS, and DUL which jointly pursue OOD detection and generalization can achieve much better overall performance compared to the baselines in Table 1 2 and 3. The reviewer suggests comparing the proposed method with these methods.
>
> **Reply:** Thanks for this feedback. **Following this suggestion, we provide experiments on the mentioned baselines. DR-SAM achieves leading performance in OOD-G and OOD-D performance.**
>
> We provide a comparison of additional baselines, including POEM, NOPS, and NTOM. We additionally conduct experiments with SCONE and WOODS. We conduct experiments with settings aligned with the original submission. As NPOS trains the CNN backbone without the final linear classifier [1], we can only provide its OOD-D performance and leave the accuracy as N/A. In addition, DUL has not yet released its source code; we will leave the comparison with it for future work.
>
> | CIFAR-10 | FPR@95 | AUROC | ID Accuracy | OOD-G Accuracy |
> | -------- | ------ | ----- | ----------- | -------------- |
> | MSP      | 39.91  | 79.24 | 95.06       | 79.24          |
> | OE       | 14.06  | 96.42 | 95.02       | 78.77          |
> | NOPS     | 28.05  | 91.85 | N/A         | N/A            |
> | POEM     | 33.02  | 88.00 | 85.68       | 66.34          |
> | WOODS    | 20.36  | 94.39 | 90.43       | 71.47          |
> | SCONE    | 19.26  | 94.71 | 91.68       | 73.56          |
> | DR-SAM   | 12.36  | 97.04 | 95.13       | 80.50          |
>
> | CIFAR-100 | FPR@95 | AUROC | ID Accuracy | OOD-G Accuracy |
> | --------- | ------ | ----- | ----------- | -------------- |
> | MSP       | 56.68  | 79.02 | 77.26       | 36.59          |
> | OE        | 36.11  | 87.64 | 76.51       | 36.24          |
> | NOPS      | 57.22  | 81.06 | N/A         | N/A            |
> | POEM      | 40.00  | 82.78 | 68.58       | 32.48          |
> | WOODS     | 56.15  | 80.32 | 77.01       | 36.48          |
> | SCONE     | 55.28  | 81.39 | 76.58       | 36.27          |
> | DR-SAM    | 38.18  | 86.93 | 77.91       | 36.91          |
>
> We notice that POEM would degenerate the model's classification ability in both in-distribution and covariate-shifted samples, which leads to sub-optimal performance compared to the OE. POEM cannot exceed the OE when using the TIN-597 as auxiliary outliers when training on the CIFAR benchmark, as DUL also has the same report. The WOODS and SCONE, on the other hand, cannot perform well in CIFAR-100 under the traditional OOD-D setting, and also scraify the OOD-G performance.
>
> **Overall, DR-SAM achieves leading performance in OOD-G and OOD-D performance on the CIFAR benchmark under the traditional OOD-D setting.** We have updated the results in Appendix B.3.
>
> [1] OpenOOD v1.5: Enhanced Benchmark for Out-of-Distribution Detection. In arXiv, 2023.

---

> ### Author Response · Authors · 2024-11-22
> **Response to Reviewer 5Bu7 (3/3)**
>
> **Q4:** About the access of semantic-shifted data.
>
> > Similar to other training-required OOD detection methods, the proposed method also needs to access semantic-shifted data during training. This limitation widely exists in many previous OOD detection methods, but still worth noting here.
>
> **Reply:** Thanks for the feedback. Follwoing this suggestion, we have add this to the discussion of the direction of the training-based OOD-D in Appendix F. **However, we should note that this submission mainly focus on handling the detection-generalization paradox instead of the requirement for semantic-shifted data access.** Accessing semantic-shifted data is commonly adopted in OE-related studies, including [1,2,3,4,5,6]. In future work, we would try to involve data synthetic methods in our framework to alleviate the dependence on auxiliary outliers.
>
> [1] Feed two birds with one scone: Exploiting wild data for both out-of-distribution generalization and detection. ICML'23
>
> [2] Unified out-of-distribution detection: A model-specific perspective. CVPR'23
>
> [3] The Best of Both Worlds: On the Dilemma of Out-of-distribution Detection. NeurIPS'24
>
> [4] Training ood detectors in their natural habitats. ICML'22
>
> [5] Poem: Out-of-distribution detection with posterior sampling. ICML'22
>
> [6] Atom: Robustifying out-of-distribution detection using outlier mining. ECML PKDD'21
>
> **Q5:** About the illustration in Fig.3(d).
>
> > In Figure 3(d), why there is no blue area (ID data) in the figure?
>
> **Reply:** Thanks for this question. Please note that the model fine-tuned with the OOD-G method would reduce the gap between $D_{\text{ID}}$ and $D_{\text{CS}}$ on the representation space, as we discovered in Sec.3.1. As a result, the learned representation are almost overlapped with each other in Fig.3(d).
>
> **Q6:** About the formulation of OOD-G in Eq.3.
>
> > I am unsure whether the formulation of OOD generalization in Eq. 3 is correct. $D_{CS}$ is mentioned by the text above Eq.3. However, there is no such notation in the following equation. The formulation should be revised carefully and a proper citation should be provided here.
>
> **Reply:** Thanks for this valuable feedback. **Following this suggestion, we follow the representative work IRM[1] and DomainBed[2] to refine the formulation.** Specifically, we consider dataset $D_{e}=${$(x_i^e, y_i^e)$}, where $i =${$1... n_e$} collected under multiple environment $e \in \mathcal{E}$. $D_{CS}$ = {$D_{e}$:$e \in \mathcal{E}$} denotes the collection of datasets from multiple environments with potential covariant shifts. We have added the above content in our updated submission in lines 146 to 148.
>
> [1] Invariant Risk Minimization. In arXiv, 2020.
>
> [2] In Search of Lost Domain Generalization. In ICLR, 2021.
>
> **Q7:** About the visualization.
>
> > The authors post a visualization of the loss landscape in Figure 10. However, such a visualization contains limited information. The reviewer suggests comparing with SAM, OE, and the original ERM.
>
> **Reply:** Thanks for this constructive feedback. We have updated the Fig.10 and added a detailed discussion on the new visualization.
>
> In Fig.10, we demonstrate the loss of landscapes for different methods on $D_{\text{CS}}^{\text{test}}$ of CIFAR-10. OE leads the model to minima with high sharpness, making it sensitive to distribution shift, thus enhancing the model's detection ability while sacrificing the generalization performance. ERM achieves higher sharpness than SAM and DR-SAM, but it does not explicitly optimize the loss's sharpness. **DR-SAM, on the other hand, achieves lower sharpness within the specific radii but high at the outer region. This allows the model to be robust to the covariate-shifted samples while sensitive to the semantic-shifted ones.**

---

> > ### Comment · Reviewer_5Bu7 · 2024-11-22
> > **Follow up**
> >
> > I thank the author for their great efforts. After going through the other reviews and the author's initial response, some of my major concerns are not well addressed. I'd like to share some additional comments.
> >
> > - The authors claim that OOD-G methods cannot ensure lower sharpness as they focus on loss consistency across domains. I am not sure whether such claim is too strong to be correct. I am still confused why OOD-G methods like SAM or mixup can not achieve low sharpness on covariate-shifted data. The mentioned two OOD-G methods, i.e., GroupDRO and VRE-x may not be represnentative enough for all OOD-G methods.
> > - One of my major concerns, i.e., the experimental settings and dependency on data augmentation remains unanwsered.
> > - The improvement of the proposed method compared to OE on CIFAR-100 in the auhor's response seems very minor.

---

> > > ### Author Response · Authors · 2024-11-23
> > > **Further responses for Reviewer 5Bu7 (2/2)**
> > >
> > > **Q3:** About the performance of DR-SAM on CIFAR-100 compared to OE.
> > >
> > > > The improvement of the proposed method compared to OE on CIFAR-100 in the auhor's response seems very minor.
> > >
> > > **Reply:** Thanks for this question. **We would like to point out that the proposed DR-SAM achieves competitive performance on OOD-D compared to OE without sacrificing the model's generalization ability.**
> > >
> > > | ImageNet-200 | OOD-D FPR@95$\downarrow$ | OOD-G Accuracy$\uparrow$ |
> > > | -------- | -------- | -------- |
> > > | MSP     | 45.16     | 43.85     |
> > > | RMDS     | 43.30     | 43.85     |
> > > | KNN     | 43.81     | 43.85     |
> > > | OE     | 44.60     | 43.30     |
> > > | DR-SAM     | 43.12     | 46.93     |
> > >
> > > **To further address the reviewer's concern about the performance comparison between OE and DR-SAM, we presented their results on ImageNet-200.** As can be seen in the above table, on more challenging ImageNet-200, OE would sacrifice the model's generalization ability, and it cannot even exceed the post-hoc method like RMDS and KNN. In contrast, DR-SAM performs slightly better than OE in OOD-D performance, while **significantly outperforming OE by 3.6% absolute improvement in OOD-G performance** on ImageNet-200, confirming the effectiveness of our proposed method in pursuit of better detection and generalization simultaneously.

---

> > > > ### Comment · Reviewer_5Bu7 · 2024-11-24
> > > >
> > > > Thanks for your continued efforts in addressing my concerns.
> > > >
> > > > - Specifically, we adopted Gaussian blur and Gaussian noise to augment the test set of the CIFAR datasets and reported the accuracy of the mixed dataset as follows.
> > > >
> > > > What kind of augment do you use on these two datasets respectively? More general augmentation like AugMix, RandomCrop may be a better choice (to clarify, there is no need to conduct additional experiments as the discussion is ending soon).
> > > >
> > > > - SAM can achieve lower sharpness on covariate-shifted data as it explicitly optimizes the sharpness of the loss.
> > > >
> > > > So what is the correct statement instead of ``OOD-G methods cannot ensure lower sharpness as they focus on loss consistency across domains''? I found similar statement still exists in the latest revision.

---

> > > > > ### Author Response · Authors · 2024-11-24
> > > > > **Further responses for Reviewer 5Bu7**
> > > > >
> > > > > We thank the reviewer 5Bu7 for the further comments.
> > > > >
> > > > > **Q1:** About the augmentation adopted in CIFAR.
> > > > > > What kind of augment do you use on these two datasets respectively? More general augmentation like AugMix, RandomCrop may be a better choice (to clarify, there is no need to conduct additional experiments as the discussion is ending soon).
> > > > >
> > > > > **Reply:** Thanks for your follow-up feedback. To address the concern about the potential information leakage in augmentation like brightness, we consider only adopting the Gaussian Blur and Guassian noise during method comparison. **Specifically, for the test set of CIFAR-10, we employ the Gaussian Blur and Gaussian noise individually to obtain two test sets augmented by either the augmentation method. We then evaluate the model's performance on these two augmented test sets together and report the accuracy. We apply the same pipeline to the CIFAR-100.**
> > > > >
> > > > > Concretely, we individually employ one of the data augmentations to augment the images. We augment the images with severity from 1 to 5. This would result in two test sets (5 times larger than the original one (5 severities)) that are augmented by either Gaussian Blur or Gaussian noise. We evaluate the model's performance on these two augmented test sets together and report the accuracy.
> > > > >
> > > > > - Following [1], Gaussian noise augments images by adding random signals drawn from a Gaussian distribution to the image. Gaussian Blur applies a Gaussian image filter from scikit-image (skimage) to augment the image. We also provide the source code as follows.
> > > > >
> > > > > ```python=
> > > > > # ref: https://github.com/hendrycks/robustness/blob/master/ImageNet-C/create_c/make_cifar_c.py
> > > > > from skimage.filters import gaussian
> > > > > import numpy as np
> > > > >
> > > > > def gaussian_noise(x, severity=1):
> > > > >     c = [0.04, 0.06, .08, .09, .10][severity - 1]
> > > > >
> > > > >     x = np.array(x) / 255.
> > > > >     return np.clip(x + np.random.normal(size=x.shape, scale=c), 0, 1) * 255
> > > > >
> > > > > def gaussian_blur(x, severity=1):
> > > > >     c = [.4, .6, 0.7, .8, 1][severity - 1]
> > > > >
> > > > >     x = gaussian(np.array(x) / 255., sigma=c, channel_axis=-1)
> > > > >     return np.clip(x, 0, 1) * 255
> > > > >
> > > > > # a test demo for the functions
> > > > > dummy_x = np.random.rand(1, 3, 224, 224) * 255 # Batch, Channels, Height, Width
> > > > > dummy_x_gaussian_noise = gaussian_noise(dummy_x, severity=1)
> > > > > dummy_x_gaussian_blur = gaussian_blur(dummy_x, severity=2)
> > > > > ```
> > > > >
> > > > > - We appreciate the reviewer's suggestion on more general augmentation methods like AugMix and RandomCrop. **These would be better choices for enhancing or evaluating the model's generalization ability.** We are trying to combine the proposed DR-SAM with AugMix and RandomCrop. We will update the performance of DR-SAM on Tabs. 1, 2, and 3 once the experiment is completed.
> > > > >
> > > > > [1] Benchmarking neural network robustness to common corruptions and perturbations. In ICLR, 2019.
> > > > >
> > > > > **Q2:** About the OOD-G method.
> > > > > > So what is the correct statement instead of ``OOD-G methods cannot ensure lower sharpness as they focus on loss consistency across domains''? I found similar statement still exists in the latest revision.
> > > > >
> > > > > **Reply:** We now add a footnote in line 90 for clarity with the explanation reference in line 91 in the revision, and refine the claim in lines 274-279. Could you check that whether these latest modifications can address your concern? Thank you so much. Any suggestion about this point is welcome, so that we can avoid unclear claim as much as possible.

---

> > > > > ### Author Response · Authors · 2024-11-25
> > > > >
> > > > > Dear Reviewer 5Bu7,
> > > > >
> > > > > Thank you very much for your further feedback.
> > > > >
> > > > > We provided detailed responses to your questions. Specifically, we
> > > > >
> > > > > - clarify augmentation adopted in CIFAR. (Q1)
> > > > > - refine the unclear claim about the OOD-G method and update the submission accordingly. (Q2)
> > > > >
> > > > > Would you mind checking our responses and confirming whether you have any further questions?
> > > > >
> > > > > Thanks for your attention and best regards. Any comments and discussions are welcome!
> > > > >
> > > > > Authors of #10299

---

> > > > > ### Author Response · Authors · 2024-11-27
> > > > > **We employ the AugMix and RandomCrop to train DR-SAM on CIFAR10/100 datasets.**
> > > > >
> > > > > Thanks for your valuable suggesstion on data augmentation. **We employ the AugMix and RandomCrop to train DR-SAM on CIFAR10/100 datasets.**  Sepcificaly, we follow the experiment setting in Tabs. 1 and 2 for training and evaluation. We also evaluate the model using the Gaussian Blur and Gaussian noise augmented dataset mentioned previously.
> > > > >
> > > > > For AugMix, we use the pytorch implementation ([link](https://pytorch.org/vision/main/generated/torchvision.transforms.AugMix.html)), set `severity` as 1 and keep other hyper-parameter as default. We employ RandomCrop with default setting following the pytorch implementation in [here](https://pytorch.org/vision/main/generated/torchvision.transforms.RandomCrop.html).
> > > > >
> > > > > | CIFAR-10            | OOD-D FPR@95$\downarrow$ | OOD-G Accuracy$\uparrow$ | OOD-G (Gaussian) Accuracy$\uparrow$ |
> > > > > |:------------------- |:------------------------:|:------------------------:|:-----------------------------------:|
> > > > > | MSP                 |          39.91           |          79.24           |                61.89                |
> > > > > | OE                  |          14.06           |          78.77           |                61.32                |
> > > > > | MixOE               |          77.08           |          79.94           |                58.36                |
> > > > > | DR-SAM (Brightness) |        **12.36**         |        **80.50**         |                63.56                |
> > > > > | DR-SAM (AugMix)     |          21.36           |          80.27           |              **65.79**              |
> > > > > | DR-SAM (RandomCrop) |          15.46           |          80.16           |                63.25                |
> > > > >
> > > > > | CIFAR-100           | OOD-D FPR@95$\downarrow$ | OOD-G Accuracy$\uparrow$ | OOD-G (Gaussian) Accuracy$\uparrow$ |
> > > > > |:------------------- |:------------------------:|:------------------------:|:-----------------------------------:|
> > > > > | MSP                 |          56.68           |          36.59           |                39.50                |
> > > > > | OE                  |          36.11           |          36.24           |                36.61                |
> > > > > | MixOE               |          57.48           |          36.95           |                36.19                |
> > > > > | DR-SAM (Brightness) |          38.18           |          36.91           |                39.44                |
> > > > > | DR-SAM (AugMix)     |          49.52           |          35.87           |                37.68                |
> > > > > | DR-SAM (RandomCrop) |        **36.96**         |        **37.11**         |              **39.64**              |
> > > > >
> > > > > From the tables shown above, we found that **Single augmentation might be more effective than the mixing ones, while mixing augmentation would hinder the performance of OOD-D.**
> > > > >
> > > > > - **Single augmentation can enhance the model's generalization and detection ability simultaneously.** Both Brightness and RandomCrop can **enhance** the model's detection and generalization performance compared to the OE and MSP baselines. The AugMix, on the other hand, would **hinder** the model's performance on the in-distribution dataset, and cannot outperform the OE in terms of OOD-D performance.
> > > > > - **Mixed augmentation would cause the model to fail to distinguish semantic-shifted samples from the covariate-shifted ones.** We observe that MixOE would also hinder the model's detection ability, which mixes up the $D_\text{ID}^{\text{train}}$ and $D_\text{SS}^{\text{train}}$ to create a smooth transition between two distribution. The observations indicate that these augmentations would weaken the model's ability to distinguish between semantic-shifted samples and covariate-shifted samples. This is likely because the boundary between distributions becomes less defined, making it harder for the model to identify and detect semantic shifts accurately.
> > > > >
> > > > > For future work, we plan to:
> > > > > - Conduct extensive experiments on the **validation** set to **uncover the relationships between different augmentation techniques and downstream performance**;
> > > > > - **Understanding the impact of augmentation on representation learning** to provide more insights into enhancing the model's detection and generalization ability.
> > > > >
> > > > > In general, the above experiment verifies our claim that "a proper data augmentation can enhance the model's detection and generalization ability simultaneously." We will explore the effect of data augmentation further to provide more insights into how it can enhance the model's detection and generalization ability.
> > > > >
> > > > > We have updated the above experiments and discussions in Appendix B.5.
> > > > >
> > > > > Since the period of the reviewer-author discussion is extended, would you mind checking our responses and confirming whether you have any further concerns so we can clarify?

---

> > > > > > ### Comment · Reviewer_5Bu7 · 2024-12-03
> > > > > > **Final Comments**
> > > > > >
> > > > > > I appreciate the authors' response and the heavy workload they have managed during the entire rebuttal period. I have carefully re-examined the latest revision, the authors' response, and the opinions of other reviewers. My previous major concerns regarding the absence of baselines (i.e., POEM, SCONE, WOODS), technical details (dependence on data augmentation), and presentation and motivation (the claim about OOD generalization) have been partially addressed. I will increase my score from 5 to 6 to avoid blocking the acceptance of this paper.
> > > > > >
> > > > > > The main reason for raising my score is that I believe the proposed method does address a practical research problem and thus deserves to be seen by more researchers. On the other hand, my score is reserved because I still have minor concerns about the overall presentation quality.
> > > > > >
> > > > > > Although it is unlikely that the manuscript will be further revised in ICLR, I have some additional suggestions and I hope they are helpful to you:
> > > > > >
> > > > > > - The logic in lines 272-284 could be made smoother.
> > > > > > - If there is chance to do further revise, the additional comparison with the suggested baselines should be integrated into the manuscript.
> > > > > > - In my opinion, it is quite acceptable to address a practical and newly-identified problem by combining existing techniques according to Occam's Razor. However, making incorrect claims that may mislead future research is unacceptable. I encourage the authors to carefully re-examine and reorganize the sections related to OOD generalization, try to compare with more representative OOD-G methods (since you claim that your method are devised for both OOD-G and OOD-D), and rewrite the related parts in a defensive style.

---

> > > > > > > ### Author Response · Authors · 2024-12-03
> > > > > > >
> > > > > > > We appreciate your valuable feedback and recognition of our work.
> > > > > > >
> > > > > > > The suggested points will be carefully revised and incorporated into the final revision:
> > > > > > > - We will carefully refine the submission's logic to make it smoother.
> > > > > > > - We will carefully re-examine and rewirte the section related to OOD-G, and conduct experiments on more OOD-G methods (e.g., Mixup and CORAL).
> > > > > > > - All the suggested baselines will be integrated into the main text.
> > > > > > >
> > > > > > > We would like to thank Reviewer 5Bu7 again for the thoughtful feedback, which has significantly contributed to our understanding of how to enhance the quality of the submission.

---

> ### Author Response · Authors · 2024-11-23
> **Further responses for Reviewer 5Bu7 (1/2)**
>
> We thank the reviewer 5Bu7 for the further comments. We addressed all these comments. Please find the detailed responses below.
>
> **Q1:** About the sharpness of OOD-G methods.
>
> > The authors claim that OOD-G methods cannot ensure lower sharpness as they focus on loss consistency across domains. I am not sure whether such claim is too strong to be correct. I am still confused why OOD-G methods like SAM or mixup can not achieve low sharpness on covariate-shifted data. The mentioned two OOD-G methods, i.e., GroupDRO and VRE-x may not be represnentative enough for all OOD-G methods.
>
> **Reply:** Thanks for your follow-up feedback. Here, we provide a further explanation as follows.
>
> - **Generally, SAM can achieve lower sharpness on covariate-shifted data as it explicitly optimizes the sharpness of the loss.** Mixup that interpolates the sample space to maintain the region consistency in prediction, to some extent, can also minimize the sharpness **in an implicit manner**. In the submission, we mean not to make a strong claim towards these methods.
> - On the other hand, **the OOD-G methods that optimize the loss via typical optimization methods**, like SGD or Adam, would generally **lead to sub-optimal performance when converging the sharp and narrow minima** [1,2,3,4,5,6,7]. This is only related to the optimizer adopted by different OOD-G methods regardless of their proposed losses.
>
> **We apologize for the previous inexact claim and agree with the reviewer's point, once we expand the taxonomy of the OOD-G methods to include SAM and mixup.** We have refined the corresponding sentences for better clarification to avoid any controversy in the taxonomy.
>
> [1] On large-batch training for deep learning: Generalization gap and sharp minima. In ICLR, 2017.
>
> [2] Loss surfaces, mode connectivity, and fast ensembling of dnns. In NeurIPS, 2018.
>
> [3] Averaging Weights Leads to Wider Optima and Better Generalization. In arXiv, 2019.
>
> [4] Sharpness-aware minimization for efficiently improving generalization. In ICLR, 2021.
>
> [5] Computing Nonvacuous Generalization Bounds for Deep (Stochastic) Neural Networks with Many More Parameters than Training Data. In arXiv, 2017.
>
> [6] Fantastic Generalization Measures and Where to Find Them. In arXiv, 2019.
>
> [7] SWAD: Domain Generalization by Seeking Flat Minima. In NeurIPS, 2021.
>
> **Q2:** About the experimental settings.
>
> > Experimental settings. The authors claim that they deploy brightness as data augmentation (Appendix C). I have concerns about whether using brightness augmentation during training can result in information leakage from the test covariate-shifted distribution. Since all the other corruptions in CIFAR10/100-C or ImageNet-C may also alter the brightness of images. As far as I could tell, in standard OOD generalization settings, the test-time covariate-shifted distribution should be kept unknown during training. Besides, the authors seem to tune the augmentation and make such a choice as they said in lines 466-468. Thus, the dependence on manually selected augmentation is a notable limitation that makes the OOD generalization problem more like domain adaption or even a trivial problem.
> >
> > One of my major concerns, i.e., the experimental settings and dependency on data augmentation remains unanwsered.
>
> **Reply:** Sorry for the late response due to the missing copy of the prepared response to this concern. To address the reviewer's concern, we would like to clarify:
>
> - **We have no intention of tuning a specific augmentation.** During training, we chose brightness as the argumentation with the other 17 augmentations for general usage purposes. We do not tune the augmentation based on the performance of the model on the test set. In lines 494-495 (corresponding to lines 466-468 in the original submission), we aim to show that a proper data augmentation can enhance the model's detection and generalization ability simultaneously, but we **do not** use the test performance as the guiding principle to tune the augmentation. Instead, **we still follow the conventional augmentation setup during the training without any specific tuning**.
>
> - **Regarding the brightness augmentation, to further address the reviewer's concern about information leakage, we conducted an experiment on the CIFAR-10C/100C dataset without using brightness augmentation.** Specifically, we adopted Gaussian blur and Gaussian noise to augment the test set of the CIFAR datasets and reported the accuracy of the mixed dataset as follows.
>
> | CIFAR-10C | OOD-G Accuracy $\uparrow$ |
> | -- | -- |
> | MSP|61.89 |
> |OE|61.32|
> |SAM| 63.17|
> |DR-SAM | 63.56|
>
> | CIFAR-100C | OOD-G Accuracy $\uparrow$ |
> | -- | --- |
> | MSP| 39.50|
> | OE| 36.61|
> | SAM| 40.74|
> | DR-SAM| 39.44|
>
> As can be seen from the table above, DR-SAM exceeds SAM in terms of OOD-G accuracy on CIFAR-10C/100C without brightness augmentation.
>
> **We appreciate the reviewer's question, and have updated the above discussion in Appendix B.4 to avoid misunderstanding.**

---

### Official Review · Reviewer_L2A2 · 2024-10-30

**Soundness:** 3
**Presentation:** 4
**Contribution:** 3
**Rating:** 6
**Confidence:** 5

**Summary:**

The paper provides research on one of the most important topic in machine learning, which is dealing with out of distribution data. Specifically, the paper present a detection-generalization paradox that exists in current machine learning systems. The authors analyze this paradox by
providing in-depth analysis of the models trained under various paradigms, focusing on representation, logits, and loss across
different data shifts. The authors propose a new idea for breaking this paradox and support their findings with extensive experiments.

**Strengths:**

The paper delves into important aspect of the non-trivial problem in existing machine learning models. One of the most important strength of this paper is the manuscript is structured in a very proper way. The introduction section is very clear, the motivation is also clear and it is very clear what the authors are trying to do. The way in which the authors conducted the in-depth analysis of the behavior of models in the representation, logits, and loss space to show the actual detection-generalization paradox is worthy of admiration. The paper also stands out well in terms of contribution, where they have presented a new methodology "Distribution Robust Sharpness Aware Minimization" which is fairly intuitive and proving effective in maintaining the detection-generalization balance in the classification systems. The authors have provided fairly good amount of literature review and conducted extensive experiments on the available benchmark datasets, and also compared with the existing references in the field. Considering the reproducibility, the authors have provided full algorithm, and released the source codes as well.

**Weaknesses:**

There are some weaknesses associated with the paper. The paper has severe typos, and a thorough proof-read is required. For instance, covariate is written as covariant in many places. The provided source code is very hard to reproduce as it does not have a readme file, and to replicate the exact experiment is difficult. There are irrelevant and lot of details in the appendices of the related works section, which is not necessary at all. The experiments and results are promising, but it could have been done better by comparing with OOD-G methods too because there is a lack of proof indicating DR-SAM can beat existing OOD-G methods. The results are majorly focused on semantic shifts based methods only.

**Questions:**

Some questions to the authors are:

1. In line 082, Analysis on logit space: For better OOD_D method enlarges the gap of prediction confidence between D_ID and D_CS. Is this a typo? Shouldn't it be D_ID and D_SS instead of D_ID and D_CS?

2. Fig. 2 is not clear. Atleast not very explaining. Why the FPR is in the range of -ve and why the OOD-G accuracy of DR-SAM around 1.7? A delta term is used for both FPR and ACC. What is this delta? Is it the difference? It needs to be clarified both in the caption and the actual. It is difficult for a normal reader to apprehend whats going in this figure.

3. In Fig. 5 the sharpness value is larger and in 6 and 7 the sharpness value is smaller. Is this the preferred characteristics of the curves for DR_SAM? Shouldn't the sharpness value in the Fig.5d remain almost steady across the value of rho? Because this characteristic contradicts with the statement made in the line 255-256.

4. Why is the ID and OOD accuracy of DR-SAM less than that of vanilla SAM? As per the result, the gain can only be seen in the AUROC which is the metric for detecting semantic shifts. What about for the covariate shift part? Shouldn't the OOD accuracy be at least on par or better than the reference and vanilla SAM as per the claim of breaking the detection-generalization paradox? Please clarify.

5. Regarding the experimental results, why the results are not compared with recent approaches that has been studied for both detection and generalization? Also, why the comparison has not been made for the Imagenet-200 benchmarks in terms of OOD-G methods in Table 3?

---

> ### Author Response · Authors · 2024-11-22
> **Response to Reviewer L2A2 (1/3)**
>
> We thank the reviewer L2A2 for the valuable feedback. We addressed all the comments. Please kindly find the detailed responses below. Any further comments and discussions are welcomed!
>
> **Q1:** About the typos.
>
> > The paper has severe typos, and a thorough proof-read is required. For instance, covariate is written as covariant in many places.
>
> **Reply:** We apologize for the typos. Following this suggestion,  we have gone through a carefully proofread and updated the paper, highlighting the changed part in blue.
>
> **Q2:** About the README file for the source code.
>
> > The provided source code is very hard to reproduce as it does not have a readme file, and to replicate the exact experiment is difficult.
>
> **Reply:** Thanks for this constructive feedback. Following this suggestion, we have provided a detailed ReadME file in the [anonymous link](https://anonymous.4open.science/r/DR-SAM-9C89/README.md). We provide detailed guidance from the environment setup to the commands used to reproduce the experiment.
>
> **Q3:** About the related works in the appendix.
>
> > There are irrelevant and lot of details in the appendices of the related works section, which is not necessary at all.
>
> **Reply:** Thanks for this valuable feedback. **Following this suggestion, we have refined the appendix to improve its readability and quality.** We organize the methods based on their underlying methodology to provide a comprehensive understanding of different methods. For OOD-G, we discuss representative work like IRM and VRE-x and reduce the potential irrelevant discussions. For OOD-D, we reduce the discussion on the post-hoc method and focus on the OE-based method. We shorten the discussion on robust finetuning to reduce the irrelevant details.
>
> **Q4:** About the OOD-G experiment.
>
> > The experiments and results are promising, but it could have been done better by comparing with OOD-G methods too because there is a lack of proof indicating DR-SAM can beat existing OOD-G methods. The results are majorly focused on semantic shifts based methods only.
>
> **Reply:** Thanks for this valuable feedback. **Please note that the OOD-G method is trained on covariate-shifted samples with explicit regulation, which allows it to be robust to potential distribution shifts.** We have provided a comparison in Tab.1 and 2 in our original submission, which shows that DR-SAM can beat OOD-G methods in terms of OOD-D performance.
>
> The OOD-G methods can enhance the model's OOD-G performance but sacrifice its ability to identify semantic-shifted samples. As shown in Tab.1 and Tab.2, the FPR@95 and AUROC of the OOD-G approaches have dropped significantly compared with that of the pre-trained model with the post-hoc OOD-D detection method (e.g., KNN, RMDS). In addition, training with OOD-G solely would also decrease the model's ID accuracy.
>
> This submission aims to enhance the model's detection and generalization ability simultaneously. As we identified the paradox at the beginning, solely training with either OOD-G or OOD-D methods cannot ensure the model excels in both tasks, which is important for real-world applications. We provide experiment results to show that DR-SAM can resolve this paradox effectively.
>
> **Q5:** About the typo.
>
> > In line 082, Analysis on logit space: For better OOD_D method enlarges the gap of prediction confidence between D_ID and D_CS. Is this a typo? Shouldn't it be D_ID and D_SS instead of D_ID and D_CS?
>
> **Reply:** Thanks for this question. **Yes, you are correct.** It should be "The OOD-D method enlarges the gap of prediction confidence between $D_{\text{ID}}$ and $D_{\text{SS}}$ (better OOD-D) and decreases the model's prediction on $D_{\text{CS}}$ (worse OOD-G)." We have updated it in the revised submission.
>
> **Q6:** Explaining the Fig.2.
>
> > Fig. 2 is not clear. Atleast not very explaining. Why the FPR is in the range of -ve and why the OOD-G accuracy of DR-SAM around 1.7? A delta term is used for both FPR and ACC. What is this delta? Is it the difference? It needs to be clarified both in the caption and the actual. It is difficult for a normal reader to apprehend whats going in this figure.
>
> **Reply:** Thanks for this constructive feedback. **Please note that we report the $\Delta$ term of the model fine-tuned with different methods ($e.g.$, OOD-D and OOD-G) by calculating the performance difference between the pre-trained and fine-tuned models.** For example, the DR-SAM's OOD-G accuracy (around 1.7) in Fig.2(a) is calculated by the result presented in Tab.1 (80.50 (DR-SAM) - 79.24(MSP)). The FPR@95 is calculated in the same manner. We have added the explanation to Fig.2's caption in the revised submission.

---

> > ### Comment · Reviewer_L2A2 · 2024-11-23
> >
> > I would like to thank the authors for clarifying the raised concerns carefully. The authors have resolved the minor issues through a thorough proofreading. Also, the question regarding Fig. 2 is clarified.

---

> ### Author Response · Authors · 2024-11-22
> **Response to Reviewer L2A2 (2/3)**
>
> **Q7:** Explaining the Fig.5.
>
> > In Fig. 5 the sharpness value is larger and in 6 and 7 the sharpness value is smaller. Is this the preferred characteristics of the curves for DR_SAM? Shouldn't the sharpness value in the Fig.5d remain almost steady across the value of rho? Because this characteristic contradicts with the statement made in the line 255-256.
>
> **Reply:** Thanks for this feedback. **Please note that DR-SAM exhibits lower sharpness within specific regions because we prompt the optimizer to do so.** We present ablation studies in Fig. 8 (a) and (b) to examine the impact of perturbation strength ($\rho$) on the model's detection and generalization performance. **The hyperparameter $\rho$ determines the neighborhood's region that the optimizer focuses on, with larger values of $\rho$ prompting the optimizer to minimize the sharpness over a broader neighborhood.** Notably, selecting an appropriate $\rho$ (e.g., $\rho = 0.5$ for CIFAR-10) can simultaneously enhance detection and generalization performance, as evidenced by the flatness observed in Fig. 5 (d).
>
> **Q8:** About the performance of DR-SAM and vanilla SAM.
>
> > Why is the ID and OOD accuracy of DR-SAM less than that of vanilla SAM? As per the result, the gain can only be seen in the AUROC which is the metric for detecting semantic shifts. What about for the covariate-shift part? Shouldn't the OOD accuracy be at least on par or better than the reference and vanilla SAM as per the claim of breaking the detection-generalization paradox? Please clarify.
>
> **Reply:** Thanks for this valuable feedback. **Please note that the employment of auxiliary outliers would cause the degeneration of OOD-G performance, and DR-SAM can alleviate this issue.**
>
> We conducted experiments on CIFAR-100 using the same settings as those adopted in Tab 2 in our original submission.
>
> | Cifar-100 | FPR@95$\downarrow$ | ID Accuracy$\uparrow$ | OOD-G Accuracy$\uparrow$ |
> | --------- | ------------------ | --------------------- | ------------------------ |
> | MSP       | 56.68              | 77.26                 | 36.59                    |
> | OE        | 33.20              | 76.51                 | 36.24                    |
> | SAM       | 53.91              | 78.53                 | 37.20                    |
> | SAM+OE    | 55.66              | 77.62                 | 36.77                    |
> | DR-SAM    | 38.18              | 77.91                 | 36.91                    |
>
> We have the following two observations from the table above:
>
> 1. **Model training with auxiliary outliers would sacrifice the model's ID and OOD-G performance.** Simply cooperating OE with SAM can improve the model's performance, but cannot compete with vanilla SAM.
> 2. **DR-SAM alleviates the side-effect of training with outliers.** This indicates the necessity of the $\text{aug}(\cdot)$ for enhancing both detection and generalization performance.
>
> We further conduct experiments on ImageNet-200 to shows that DR-SAM outperform the vanilla SAM in both OOD-D and OOD-G, shown as table follows.
>
> | ImageNet-200 | FPR@95$\downarrow$ | OOD-G Accuracy$\uparrow$ |
> | ------------ | ------------------ | ------------------------ |
> | MSP          | 56.68              | 43.85                    |
> | SAM          | 46.31              | 42.71                    |
> | DR-SAM       | 43.12              | 46.93                    |
>
> In general, DR-SAM effectively resolves the detection-generalization paradox.

---

> > ### Comment · Reviewer_L2A2 · 2024-11-23
> >
> > I thank the authors for clarifying the questions. However, I am still not fully convinced with some answers and have some follow up questions.
> > Q7) Thank you for the answers. But the question was really about steady performance (desired) of DR-SAM across the range of rho where the values of FPR@95 should remain low and OOD-G acc should remain high. Ideally, we are aiming here to achieve lower sharpness value which should equate to lower FPR@95 and higher OOD-G acc, however, this characteristics is not observed for DR-SAM in Fig 5d. Instead, the pretrained model is showing low and steady sharpness across range of rhos.
> >
> > Q8) Thanks for the reply. I can understand why the SAM+OE can beat vanilla SAM, because of its capability to introduce extra regularization with outliers for dealing with semantic shifted samples. And the degeneration of OOD-G with SAM+OE is understandable. However, my concern here is about DR-SAM (which does not use OE) should atleast have FPR@95 and OOD-G accuracy better than the vanilla SAM. I see that the authors have updated the new results for Imagenet-200, whereas for CIFAR-100 case, according to the table SAM is still better than DR-SAM across all aspects.

---

> ### Author Response · Authors · 2024-11-22
> **Response to Reviewer L2A2 (3/3)**
>
> **Q9:** About the experimental results.
>
> > Regarding the experimental results, why the results are not compared with recent approaches that has been studied for both detection and generalization? Also, why the comparison has not been made for the Imagenet-200 benchmarks in terms of OOD-G methods in Table 3?
>
> **Reply:** Thanks for this feedback. **Following this suggestion, we conduct experiments on additional baselines, including WOODS and SCONE, with settings aligned with the original submission.** We then provide justification for not adopting OOD-G methods on ImageNet-200 benchmarks.
>
> - **Overall, DR-SAM achieved leading performance in OOD-G and OOD-D performance on the CIFAR benchmark under the traditional OOD-D setting.**
>
> | CIFAR-10 | FPR@95 | AUROC | ID Accuracy | OOD-G Accuracy |
> | -------- | ------ | ----- | ----------- | -------------- |
> | MSP      | 39.91  | 79.24 | 95.06       | 79.24          |
> | OE       | 14.06  | 96.42 | 95.02       | 78.77          |
> | WOODS    | 20.36  | 94.39 | 90.43       | 71.47          |
> | SCONE    | 19.26  | 94.71 | 91.68       | 73.56          |
> | DR-SAM   | 12.36  | 97.04 | 95.13       | 80.50          |
>
> | CIFAR-100 | FPR@95 | AUROC | ID Accuracy | OOD-G Accuracy |
> | --------- | ------ | ----- | ----------- | -------------- |
> | MSP       | 56.68  | 79.02 | 77.26       | 36.59          |
> | OE        | 36.11  | 87.64 | 76.51       | 36.24          |
> | WOODS     | 56.15  | 80.32 | 77.01       | 36.48          |
> | SCONE     | 55.28  | 81.39 | 76.58       | 36.27          |
> | DR-SAM    | 38.18  | 86.93 | 77.91       | 36.91          |
>
> As can be seen from the table above, WOODS and SCONE cannot perform well in CIFAR-100 under the traditional OOD-D setting, and also scraify the OOD-G performance. We have updated the results in Appendix B.3.
>
> - **We follow the common setting adopted in [1,2,3,4,5] to conduct OOD-G experiments.** Generally, OOD-G methods focus on small-scale datasets like the CIFAR benchmark but on large-scale ones like ImageNet-200. Since we aim to evaluate the model's detection and generalization performance, we conduct OOD-G experiments on the commonly adopted CIFAR benchmark. The scale and difficulty of the ImageNet-200 are much larger and harder than the CIFAR benchmark, which requires a longer optimization time. We are running the experiment and will update once we have the results. We will update the ImageNet-200 experiments to Tab.3 in the main text.
>
> [1] Towards out-of-distribution generalization: A survey. In arXiv, 2021.
>
> [2] Invariant Risk Minimization. In arXiv, 2020.
>
> [3] Out-of-Distribution Generalization via Risk Extrapolation. In ICML, 2021.
>
> [4] In Search of Lost Domain Generalization. In ICLR, 2021.
>
> [5] Feed Two Birds with One Scone: Exploiting Wild Data for Both Out-of-Distribution Generalization and Detection. In ICML, 2023.

---

> > ### Comment · Reviewer_L2A2 · 2024-11-23
> >
> > I thank the authors for providing response to the answers of my questions. The authors have fully clarified on why Imagenet-200 is not added in the experiment at the moment and it is totally understandable. Also, to make the comparison fully fair, the authors have introduced the results of new references WOODS and SCONE. And based on the provided table, it is clear that DR-SAM can provide good performance beating these methods.

---

> ### Author Response · Authors · 2024-11-23
> **Further response for Reviewer L2A2**
>
> We thank the reviewer L2A2 for the further comments. We addressed all these comments. Please find the detailed responses below.
>
> **Q1:** About the sharpness of the model.
>
> > Q7) Thank you for the answers. But the question was really about steady performance (desired) of DR-SAM across the range of rho where the values of FPR@95 should remain low and OOD-G acc should remain high. Ideally, we are aiming here to achieve lower sharpness value which should equate to lower FPR@95 and higher OOD-G acc, however, this characteristics is not observed for DR-SAM in Fig 5d. Instead, the pretrained model is showing low and steady sharpness across range of rhos.
>
> **Reply:** Thanks for this question. We would like to answer this question in threefold. The $\rho$ here not only indicate the neighborhood radii, but also act as the hyper-parameter for training the DR-SAM.
>
> - **DR-SAM would have better and steady OOD-D and OOD-G performance when optimized with lower $\rho$.** As shown in Fig. 8 (a) and (b), a smaller $\rho$ ($\rho$=0.5) is desired to have better detection performance. We also notice that $\rho$ has a relatively weak impact on the model's OOD-G performance within the specific region ($\rho \in [0,1.5]$). Thus, to enhance the detection and generalization performance of the model simultaneously, we choose $\rho=0.5$.
> - **The sharpness is only related to the OOD-G performance of the model.** Lower sharpness indicates higher OOD-G accuracy [1], as we discussed in our original submission in lines 256 to 257. The sharpness shows less of a relationship with the detection performance of the model.
> - **A steady low sharpness across range of rhos on $D_{\text{ID}}^{\text{train}}$ might not ensure low sharpness on $D_{\text{ID}}^{\text{test}}$ (Fig. 6 (d)) and $D_{\text{CS}}^{\text{test}}$(Fig. 7 (d)).** Compared to DR-SAM, the pretrained and OE fine-tuned model shows steady low sharpness on $D_{\text{ID}}^{\text{train}}$ (Fig. 5 (d)) but higher sharpness on $D_{\text{CS}}^{\text{test}}$ (Fig. 7 (d)). As a result, their generalization performance cannot exceed the OOD-G methods.
>
> In general, DR-SAM would have better performance when $\rho$ is smaller ($\rho=0.5$). A steady low sharpness across range of $\rho$ on $D_{\text{ID}}^{\text{train}}$ might not leads to better generlization on $D_{\text{CS}}^{\text{test}}$.
>
> [1] Sharpness-aware minimization for efficiently improving generalization. In ICLR, 2021.
>
> **Q2:** About the performance of DR-SAM and vanilla SAM on CIFAR-100.
>
> > Q8) Thanks for the reply. I can understand why the SAM+OE can beat vanilla SAM, because of its capability to introduce extra regularization with outliers for dealing with semantic shifted samples. And the degeneration of OOD-G with SAM+OE is understandable. However, my concern here is about DR-SAM (which does not use OE) should atleast have FPR@95 and OOD-G accuracy better than the vanilla SAM. I see that the authors have updated the new results for Imagenet-200, whereas for CIFAR-100 case, according to the table SAM is still better than DR-SAM across all aspects.
>
> **Reply:** Thanks for this question. We would like to respectfully clarify the misunderstanding of the reviewer regarding the FPR@95 metric and DR-SAM. **Actually, our DR-SAM does employ OE. Regarding performance, DR-SAM is better than SAM in terms of OOD-D metric FPR@95 (lower is better), and achieves competitive OOD-G accuracy compared to SAM on CIFAR-100.**
>
> - **DR-SAM employs OE to enhance the model's detection ability.** In Algorithm 1 of the submission, we show that DR-SAM employs auxiliary outliers to enhance the model's detection ability. By conducting experiments on SAM+OE, We show that without $aug(\cdot)$, simply cooperating SAM with OE cannot enhance the detection and generalization ability simultaneously. The above arguments further justify the vadality of the proposed DR-SAM in handling the detection-generalization paradox.
>
> | CIFAR-100 | OOD-D FPR@95$\downarrow$ | OOD-G Accuracy$\uparrow$ |
> | -------- | -------- | -------- |
> | MSP     | 56.68     | 36.59     |
> | OE     | 36.11     | 36.24     |
> | SAM     | 53.91     | 37.20     |
> | DR-SAM     | 38.18     | 36.91     |
>
> - **DR-SAM exceeds the OOD-D performance (FPR@95) of SAM by a large margin.** As the table above shows, DR-SAM alleviates the issue of training with OE by improving the model's generalization ability (OOD-G accuracy). We should also note that as our goal is to achieve good detection and generalization performance simultaneously, the results should be **considered collectively in both measures**.
>
> **In general, DR-SAM enhances the model's generalization and detection ability by training with $aug(D^{\text{train}}_{ID})$ and auxiliary outliers (OE).**

---

> > ### Comment · Reviewer_L2A2 · 2024-11-25
> >
> > Thank you authors for the clarification on the additional questions. However, it is highly suggested to reflect them in the manuscript as well for better understanding to the readers.

---

> > > ### Author Response · Authors · 2024-11-25
> > > **Further response for Reviewer L2A2**
> > >
> > > We thank the reviewer L2A2 for the valuable suggestions.
> > >
> > > - We have updated the Q1 to lines 280 to 282 in Sec.3.2, and refer to Appendix C for detailed discussion.
> > > - We also provide a detailed comparison of DR-SAM, standard SAM, and SAM+OE in Appendix A.
> > >
> > > We kindly refer the reviewer to check our latest version of the submission. Any further suggestions and discussions are welcome!

---

> > > > ### Comment · Reviewer_L2A2 · 2024-12-01
> > > > **Final Decision after author's response**
> > > >
> > > > I would like to thank the authors for fully engaging in this discussion and trying to answer the questions raised throughout the review process. The paper is written well, and after multiple rebuttals the quality of the paper seems to be raised in terms of soundness and technicality. The topic is also very interesting indeed and it is also interesting how the authors have tried to see the distribution shift from both aspects ; semantic and covariate. The findings provide new insights into seeing the OOD problem and hopefully impact the upcoming research area in this domain. Therefore, considering the latest version of the paper, I change my decision to accept.

---

> > > > > ### Author Response · Authors · 2024-12-03
> > > > >
> > > > > We appreciate your thorough feedback and the recognition of our work. Thank you for helping us improve the submission.

---

### Official Review · Reviewer_iCQm · 2024-11-03

**Soundness:** 2
**Presentation:** 2
**Contribution:** 2
**Rating:** 5
**Confidence:** 2

**Summary:**

This paper introduces the concept of the Detection-Generalization Paradox and analyzes the detailed reasons why existing OOD-D and OOD-G methods lead to this phenomenon. It proposes DR-SAM to simultaneously enhance the model's detection and generalization capabilities for OOD data.

**Strengths:**

1. This paper decomposes the inference process and provides a detailed analysis of the reasons behind the Detection-Generalization Paradox.

2. This paper validate the phenomenon of the Detection-Generalization Paradox from the perspectives of landscape and sharpness.

3. Experimental results demonstrate that DR-SAM simultaneously enhances the performances of OOD-D and OOD-G ability.

**Weaknesses:**

1. The proposed method DR-SAM lacks innovation. It appears to combine OE and SAM as optimization objectives, with an additional data augmentation to calculate perturbation factor $\epsilon$.

2. The analysis of the method is not detailed enough. In Algorithm 1, should $f_{\theta+\epsilon}$ in lines 3 be $f_{\theta}$?

3. In Algorithm 1, does using data augmentation to calculate the perturbation factor $\epsilon$ in line 4 affect the model's performance on $D_{ID}^{test}$ compared to vanilla SAM?

4. Is the capability for OOD-G derived from data augmentation or SAM? The authors should include relevant ablation experiments to clarify this.

5. Data augmentation seems to be the most significant innovation in DR-SAM, and the authors should include experiments to demonstrate the impact of having or not having data augmentation.

**Questions:**

Please see the weaknesses.

---

> ### Author Response · Authors · 2024-11-22
> **Response to Reviewer iCQm (1/3)**
>
> We thank the reviewer, iCQm, for the valuable feedback. We addressed all the comments. Please kindly find the detailed responses below. Any further comments and discussions are welcomed!
>
> **Q1:** About the contribution.
>
> > The proposed method DR-SAM lacks innovation. It appears to combine OE and SAM as optimization objectives, with an additional data augmentation to calculate perturbation factor $\epsilon$.
>
> **Reply:** Thanks for this comment. We would kindly note that **simply combining OE and SAM without $\text{aug}(\cdot)$ cannot simultaneously enhance detection and generalization performance.** Here, we first clarify the difference between DR-SAM and standard SAM. Second, we show that simply combining SAM with OE cannot handle the detection-generalization paradox. Finally, we highlight our contribution.
>
> - **A key difference between DR-SAM and the standard SAM is employing $\text{aug}(\cdot)$ only during the perturbation generation phase.** Analytically, we compare the difference between DR-SAM and standard SAM in the following table from three: (1) the ultimate optimization target, (2) the data sources for acquiring perturbation, and (3) the parameter optimization stage.
>
> |                       | SAM                                                          | DR-SAM                                                       |
> | --------------------- | ------------------------------------------------------------ | ------------------------------------------------------------ |
> | Optimization targets  | Enhance model's generalization ability on $D_{ID}$.          | Enhance the model's generalization ability on $D_{CS}$ while optimizing with auxiliary outliers to boost the detection ability on $D_{SS}$, **which effectively handles the detection-generalization paradox.** |
> | Acquire perturbation  | $D^{\text{train}}_{ID}$ (**only pursue low sharpness in $D_{ID}^{\text{train}}$.** It does not explicitly involve either covariate-shifted or semantic-shifted samples, and cannot guarantee the model's detection performance under fine-grained datasets like CIFAR-100 or ImageNet-200.) | $aug(D_{ID}^{\text{train}})$ with auxiliary outliers (**pursue low sharpness in both $D_{ID}^{\text{train}}$ and $aug(D_{ID}^{\text{train}})$.** DR-SAM obtains perturbation using **both** simulated covariate-shifted samples and auxiliary semantic-shifted data, to create a challenging perturbation aware of the worst case for generalization under covariate shift, while exploring the separation between $D_{ID}$ and $D_{SS}$. |
> | Gradient optimization | Perform gradient descent using gradient signal from the perturbed point. **This process only guarantees the model's training trajectory for ID classification.** | Perform gradient descent using gradient signal from the perturbed point. This process involves samples from $D^{\text{train}}_{ID}$ and auxiliary outliers, allowing us to preserve the benefits of the training trajectory for OOD-D. **This process enables us to improve generalization and detection simultaneously in a more harmonious manner.** |

---

> ### Author Response · Authors · 2024-11-22
> **Response to Reviewer iCQm (2/3)**
>
> - **Simple "SAM+OE" cannot enhance detection and generalization simultaneously.** To further justify the validity of $\text{aug}(\cdot)$, we simply cooperate auxiliary outliers to obtain perturbation for SAM optimization, termed as "SAM+OE." We compare the SAM+OE with standard SAM and proposed DR-SAM as follows:
>
> | Cifar-10 | FPR@95$\downarrow$ | AUROC$\uparrow$ | ID Accuracy$\uparrow$ | OOD-G Accuracy$\uparrow$ |
> | -------- | ------------------ | --------------- | --------------------- | ------------------------ |
> | SAM      | 25.15              | 92.08           | 95.69                 | 80.69                    |
> | SAM+OE   | 30.95              | 90.94           | 95.27                 | 80.24                    |
> | DR-SAM   | 12.36              | 97.04           | 95.13                 | 80.50                    |
>
>
> | Cifar-100 | FPR@95$\downarrow$ | AUROC$\uparrow$ | ID Accuracy$\uparrow$ | OOD-G Accuracy$\uparrow$ |
> | --------- | ------------------ | --------------- | --------------------- | ------------------------ |
> | SAM       | 53.91              | 80.14           | 78.53                 | 37.20                    |
> | SAM+OE    | 55.66              | 79.56           | 77.62                 | 36.77                    |
> | DR-SAM    | 38.18              | 86.93           | 77.91                 | 36.91                    |
>
> As can be seen from the tables, **simply combining SAM+OE cannot enhance both targets simultaneously**, indicating the validity of introducing the $\text{aug}(\cdot)$ for creating a challenging perturbation aware of the worst case for generalization under covariate shift.
>
> - **The proposed DR-SAM is based on the previous analysis of the detection-generalization paradox we identified.** Specifically, we identify the detection-generalization paradox, arising from the prevailing OOD-D and OOD-G methods. We analyze the reason behind the paradox based on three informative variables: representation, logit, and the loss of space with $D_{ID}, D_{CS},$ and $D_{SS}$. We reveal that an ideal model should have a flatter neighborhood around the converging area ($i.e.,$ low sharpness) on both $D_{ID}, D_{CS}$ data. This motivated us to develop the DR-SAM to search for such an ideal model.
>
> **In general, the comparison and experiments justify DR-SAM's novelty and the necessity of $\text{aug}(\cdot)$, as simply cooperating SAM with OE cannot resolve the detection-generalization paradox.**
>
> We have added the above comparison and experiments to the Appendix A. We kindly refer the reviewer to check the latest version of our submission.
>
> **Q2:** About the typos.
>
> > The analysis of the method is not detailed enough. In Algorithm 1, should $f_\theta + \epsilon$ in lines 3 be $f_\theta$?
>
> **Reply:** Thanks for this valuable feedback. **Yes, it should be $f_\theta$ in lines 3.** We would like to explain the method by describing it in three steps as follows:
>
> 1. **Acquire perturbation**: We utilize the augmented samples with auxiliary outliers to enhance the model's ability to identify covariant-shifted samples from semantic-shifted ones.
> 2. **Calculate the gradient**:  We then pose the perturbation to the model and calculate the gradient using in-distribution samples without aug(·) and outliers to preserve the benefits of the training trajectory for OOD-D.
> 3. **Update model's parameters**: We update the model's unperturbed parameters using the gradient signal from the previous step to improve generalization and detection more harmoniously.
>
> The analysis in Sec.3 indicates that an ideal model should possess low sharpness on both $D_{ID}$ and $D_{CS}$ data. We thereby propose DR-SAM to enhance the model's detection and generalization ability simultaneously. **The pipeline illustrated above allows us to create a challenging perturbation aware of the worst case for generalization under covariate shift while exploring the separation between $D_{ID}$ and $D_{SS}$.**

---

> ### Author Response · Authors · 2024-11-22
> **Response to Reviewer iCQm (3/3)**
>
> **Q3:** About the effect of data augmentation.
>
> > In Algorithm 1, does using data augmentation to calculate the perturbation factor $\epsilon$ in line 4 affect the model's performance on $D^{\text{test}}_{\text{ID}}$ compared to vanilla SAM?
>
> **Reply:** Thanks for this valuable feedback. **The employment of auxiliary outliers cause the degeneration on $D^{\text{test}}_{\text{ID}}$, and data augmentation can alleviate this issue.**
>
> We conducted experiments on CIFAR-100 using the same settings as those adopted in Tab 2 in our original submission.
>
> | Cifar-100 | FPR@95$\downarrow$ | ID Accuracy$\uparrow$ |
> | --------- | ------------------ | --------------------- |
> | MSP       | 56.68              | 77.26                 |
> | OE        | 33.20              | 76.51                 |
> | SAM       | 53.91              | 78.53                 |
> | SAM+OE    | 55.66              | 77.62                 |
> | DR-SAM    | 38.18              | 77.91                 |
>
> We have the following two observations from the table above:
>
> 1. **Model training with auxiliary outliers would sacrifice the model's ID performance.** Simply cooperating OE with SAM can improve the model's ID performance, but cannot compete with vanilla SAM. In addition, SAM+OE also fails to achieve competitive ID performance with vanilla SAM.
> 2. **DR-SAM alleviates the side-effect of training with outliers.** This indicates the necessity of the $\text{aug}(\cdot)$ for enhancing both detection and generalization performance.
>
> In general, DR-SAM effectively resolves the detection-generalization paradox.
>
> **Q4:** Ablation study on data augmentation.
>
> > Is the capability for OOD-G derived from data augmentation or SAM? The authors should include relevant ablation experiments to clarify this.
> >
> > Data augmentation seems to be the most significant innovation in DR-SAM, and the authors should include experiments to demonstrate the impact of having or not having data augmentation.
>
> **Reply:** Thanks for this valuable suggestion. **Following this suggestion, we empirically show that the capability for OOD-G of DR-SAM is derived from data augmentation.** We conduct ablation experiments on CIFAR-10 and ImageNet-200.
>
> | CIFAR-10                         | FPR@95$\downarrow$ | OOD-G Accuracy$\uparrow$ |
> | -------------------------------- | ------------------ | ------------------------ |
> | MSP                              | 39.91              | 79.24                    |
> | SAM                              | 25.15              | 80.69                    |
> | DR-SAM                           | 12.36              | 80.50                    |
> | DR-SAM (w/o $\text{aug}(\cdot)$) | 15.46              | 80.16                    |
>
> | ImageNet-200                     | FPR@95$\downarrow$ | OOD-G Accuracy$\uparrow$ |
> | -------------------------------- | ------------------ | ------------------------ |
> | MSP                              | 56.68              | 43.85                    |
> | SAM                              | 46.31              | 42.71                    |
> | DR-SAM                           | 43.12              | 46.93                    |
> | DR-SAM (w/o $\text{aug}(\cdot)$) | 45.10              | 44.73                    |
>
> As can be seen from the table above, compared to the MSP baseline, vanilla SAM can enhance the model's OOD-G performance in CIFAR-10 but fails in more fine-grained datasets like ImageNet-200. Whereas DR-SAM (w/o $\text{aug}(\cdot)$) would degenerate the model's OOD-G and OOD-D performance compared to the DR-SAM. **The above experiments indicate that the capability for OOD-G is derived from data augmentation rather than SAM.**
>
> We also update the results in our revised submission in Appendix B. We kindly refer the reviewer to check our latest version of the submission.

---

> > ### Author Response · Authors · 2024-11-25
> > **Would you mind checking our responses and confirming whether you have any further questions?**
> >
> > Dear Reviewer iCQm,
> > Thanks very much for your time and valuable comments.
> >
> > We understand you might be quite busy. Would you mind checking our responses and confirming whether you have any further questions?
> >
> > Any comments and discussions are welcome!
> >
> > Thanks for your attention and best regards.
> >
> > Authors of #10299

---

> > ### Comment · Reviewer_iCQm · 2024-12-01
> >
> > Thanks for the response. Most of my concerns have been addressed, but I still think the novelty of combining OE and SAM is limited. I will keep my rating.

---

> > > ### Author Response · Authors · 2024-12-03
> > > **A further response to Reviewer iCQm**
> > >
> > > **Q:** About the novelty.
> > >
> > > > Thanks for the response. Most of my concerns have been addressed, but I still think the novelty of combining OE and SAM is limited. I will keep my rating.
> > >
> > > **Reply:** Thanks for this feedback. **Please note that DR-SAM does not simply combine OE and SAM.**  In addition, **[Reviewer L2A2](https://openreview.net/forum?id=w0jk3L3IjV&noteId=6dKjaXPd1z) and [Reviewer 5Bu7](https://openreview.net/forum?id=w0jk3L3IjV&noteId=uaGWDGb7cC) have recognized our technical contribution to the important problem** (detection-generalization paradox), **which deserves to be seen by the community.**
> > >
> > > Besides, we have provided **detailed** experiments and discussions in [Q1](https://openreview.net/forum?id=w0jk3L3IjV&noteId=ONjv4bZFZ5) of our initial responses to show that:
> > > 1. **Simply combining OE with SAM cannot enhance detection and generalization simultaneously.** The SAM+OE would even underperform the vanilla SAM (up to **18.73%** relative decreasment in OOD-D performance). Our DR-SAM does not simply combine OE and SAM. The effectiveness of DR-SAM is recognized by Reviewer [Reviewer L2A2](https://openreview.net/forum?id=w0jk3L3IjV&noteId=6dKjaXPd1z) and [Reviewer 5Bu7](https://openreview.net/forum?id=w0jk3L3IjV&noteId=uaGWDGb7cC).
> > > 2. **DR-SAM differs from vanilla SAM in three ways: optimization target, perturbation acquisition, and gradient optimization.** Specifically, DR-SAM distinguishes itself by acquiring perturbation from $aug(D^{\text{train}}_{ID})$ with auxiliary outliers, which can enhance the model's generalization and detection performance. This also justifies the technical contribution of DR-SAM.
> > > 3. **We propose DR-SAM based on the analysis of the detection-generalization paradox.** We analyze the paradox based on three informative variables: representation, logit, and the loss of space with $D_{ID}, D_{CS},$ and $D_{SS}$. The insights revealed by the analysis motivated us to develop the DR-SAM to handle the problem. This contribution is also recognized by [Reviewer L2A2](https://openreview.net/forum?id=w0jk3L3IjV&noteId=6dKjaXPd1z).
> > >
> > > **Therefore, we believe that the above discussions, experiments, together with the recognition from other reviewers, can justify our technical contribution to the important research problem.**

---

> ### Author Response · Authors · 2024-11-24
> **Would you mind checking our responses and confirming whether you have any further questions?**
>
> Dear Reviewer iCQm,
>
> Thank you very much for your time and valuable comments.
>
> In the rebuttal period, we provided detailed responses to all your comments and questions point-by-point. Specifically, we
>
> - clarify the difference between DR-SAM and standard SAM. (Q1)
> - amend typos and explain our method further. (Q2)
> - explain the effect of data augmentation on $D_\text{ID}^{\text{test}}$. (Q3)
> - conduct ablation studies on data augmentation. (Q4)
>
> Would you mind checking our responses and confirming whether you have any further questions?
>
> Any comments and discussions are welcome!
>
> Thanks for your attention and best regards.

---

> ### Comment · Area_Chair_ypcm · 2024-12-01
> **Reminder**
>
> Dear Reviewer iCQm,
>
> The authors have submitted their rebuttal. Could you please take a moment to review their response and confirm whether your concerns have been addressed? Thank you for your efforts and contributions.
>
> Best regards,
>
> Your Area Chair

---

### Author Response · Authors · 2024-11-22
**A General Response by Authors**

We would like to thank all the reviewers for their valuable comments on our work.

We are glad that all the reviewers have generally good impressions of our work, including
- Important and non-trival problem (L2A2, 5Bu7);
- In-depth analysis (iCQm, L2A2);
- Well-structured paper and good writing (L2A2, 5Bu7);
- Effective experimental results (iCQm, L2A2);

**In the rebuttal period, we have provided detailed responses to all the comments and questions point-by-point.** Specifically,
- Clarify the setting (Q3 for iCQm; Q4, Q8 for L2A2; Q4, Q5 for 5Bu7;), contribution (Q1 for iCQm; Q1 for 5Bu7; ), and motivation (Q2 for 5Bu7; );
- Additional experiments (Q3, Q4 for iCQm; Q9 for L2A2; Q3 for 5Bu7; Q8 for 5Bu7);
- Explain the figures (Q6, Q7 for L2A2);
- Amende typo (Q2 for iCQm; Q1, Q5 for L2A2; Q6, Q7 for 5Bu7);
- Add instructions on reproducing the code (Q2 for L2A2);
- Reorganize the related works in the appendix. (Q3 for L2A2);

Lastly, we would appreciate all reviewers’ time again. Would you mind checking our response and confirming whether you have any further questions? **We are anticipating your feedback!**

---

### Comment · Area_Chair_ypcm · 2024-11-30

Dear Reviewers,

The public discussion phase is ending soon (2nd Dec). Active participation is highly appreciated and recommended. Thanks for your efforts and contributions.

Best regards,

Your Area Chair

---

### Meta-Review · Area_Chair_ypcm · 2024-12-17

**Metareview:**

This paper proposes a novel optimization framework for both OOD generalization and OOD detection. The method is motivated by the conflict between OOD generalization and detection, which is a novel research problem. The authors first analyze the behaviors of models trained with different paradigms in terms of loss landscape. This motivates the proposed DR-SAM to encourage low sharpness on both ID and covariate-shifted data. All reviewers agreed the method addresses an important research problem. However, more than one reviewer found the writing issues make this manuscript difficult to follow, with severe typos in the initial version, and thorough proofreading is required. Some felt a few important baselines were missing, and some claims in this paper need to be corrected. The authors addressed many concerns presented and agreed to address the presentation issues. At the end of the rebuttal, there were neither unanimous reviews nor strong recommendations for acceptance. AC gave a closer look at this paper and felt the research problem is important and the proposed method is novel. However, as the presentation issues would significantly limit the impact of the paper, the overall changes required for this paper may be extensive, which necessitates further review before its publication on ICLR.

**Additional Comments On Reviewer Discussion:**

Two of the three reviewers raised concerns about the presentation quality, lack of comparison with recent works, and technical/experimental details. The authors address many concerns. One reviewer did not actively participate in the discussion and was of low confidence. Thus, the rating should be of low weight. Since the overall changes required for this paper are extensive, it can not be accepted by ICLR in its current version.

---

### Decision · Program_Chairs · 2025-01-22

Reject